# Developmental constraints in the repeated evolution of male tail characters in rhabditid and diplogastrid nematodes

**Karin Kiontke, Simone Kolysh, Rocio Ng, David H. A. Fitch**[ORCID]*

Center for Developmental Genetics, Department of Biology, New York University, New York, New York, United States of America

* david.fitch@nyu.edu

## Abstract

A longstanding question in evolutionary biology is how change might be restricted or biased due to developmental constraints. To address this question, we investigated three recurrently evolving characters in rhabditid nematode male tails: tail tip morphogenesis, the number of genital papillae (GPs or "rays"), and phasmid position relative to the three most posterior GPs. This new analysis incorporates taxa (rhabditids *Cruznema tripartitum*, *Haematozoon subulatum*, *Poikilolaimus oxycercus*, diplogastrid *Diplogasteroides nasuensis*, and outgroup representative *Brevibucca saprophaga*) representing more and deeper divergence points in the rhabditid phylogeny than in previous analyses, allowing better resolution of ancestral states and changes. Analysis of GP characters was accomplished via immunofluorescent staining of adherens junctions at different stages of GP development and laser microbeam ablations of GP primordia. Findings include the following: (1) Loss and gain of tail tip morphogenesis occurred multiple times, possibly involving differences in fusion. (2) The pattern of GP anlagen in early L4 males is highly conserved and compatible with the previously proposed "archetype," but is established at different developmental times in different species, consistent with constraint on GP patterning by the cell lineage and antero-posterior and dorsoventral patterning systems. (3) The stem species of Rhabditina likely had 8 GPs, with the second GP (v2) gained after the divergence of *Poikilolaimus*; within rhabditids, a different GP (v6) appears to be lost twice independently. (4) Laser ablation showed that changes in phasmid position relative to GPs are not due to changes in cell lineage, but instead due to migratory switches in the relative positions of precursors of phasmid socket cells and GPs; these cell migrations occur at different developmental times in different species. In summary, our results indicate a strong constraint imposed on the cell lineage and dorsoventral positioning of GP precursors, with GP pattern diversity allowed by cell-specific migratory behavior.

**Data availability statement:** All relevant data are within the manuscript and its Supporting Information files.

**Funding:** This work was funded by grants to DHAF from the National Institutes of Health (R01-GM141395) and the National Science Foundation (IOB-0643047). SK and RN were supported by Deans Undergraduate Research Fellowships from the College of Arts and Science at New York University. The funders had no role in study design, data collection and analysis, decision to publish, or preparation of the manuscript.

**Competing interests:** The authors have declared that no competing interests exist.

## Introduction

A major question in evolutionary biology is how often the same developmental events, mechanisms and genes are reused to make similar phenotypes [1–5]. That is, how strongly do developmental genetic mechanisms constrain the evolutionary pathways leading to morphologies? To address this question, one can exploit the natural "evolutionary experiments" that have occurred during "homoplasy," i.e., the recurrent evolution of similar morphological features in independent phylogenetic lineages. In the absence of strong developmental constraints, chances are that homoplastic changes occur via different developmental mechanisms (i.e., "convergence"). On the other hand, if the same mechanism is involved each time (i.e., "parallelism"), developmental constraint is indicated. Some investigators have proposed that parallelism is expected to be quite common due to deep conservation of gene regulatory networks and the transcription factors that act at the hubs of these networks [3,6]; many examples have been found (e.g., [7–10]). Thus, because stronger constraints would result in the same mechanisms being employed recurrently, we use parallelism here as an operational indicator for constraint.

Here, we investigate origins of homoplasy by characterizing key developmental events producing male tail characters in rhabditid nematodes that show such homoplasy. These characters have traditionally been important in nematode systematics. The three characters we investigate here involve tail tip morphology, numbers and positions of genital papillae (GPs), and the position of the phasmid relative to these GPs [11–13]. In different species, the shape of the adult tail tip can be long and pointed (leptoderan, Lep, as in *Panagrellus redivivus*, Fig 1B) or short and round (peloderan, Pel, as in *C. elegans*, Fig 1A). Tail tips in juvenile stages of both sexes of most species are pointed, and the tails of Lep males retain this juvenile shape; however, in species with Pel males, the tail tip cells undergo the process of tail tip morphogenesis (TTM) in which the tail tip cells round up and retract anteriorly [13]. Males also have a bilaterally paired series of mechanosensory sensilla surrounding the cloaca called genital papillae (GPs, "rays" in *C. elegans*). In rhabditids, there are typically 9 pairs of GPs, but some species have only 8. Finally, males possess a pair of chemosensory phasmids that are also present in females. In some species, the phasmids in males can also take the shape of papillae, and have sometimes been mistaken for GPs [11,12]. The position of the phasmids relative to the GPs varies, with nearly all variation falling into two discrete types: (1) phasmids are posterior to all GPs or (2) are anterior to the most posterior three GPs [12].

The first step in analyzing recurrent evolution is to determine how often and in what species lineages the evolutionary changes occurred by mapping character states on a phylogenetic tree of the species in question. Once independent changes are identified, a second step involves looking for clues in the development of these characters that would indicate constraints or the lack thereof. That is, we can examine developmental events such as cell fusions, migrations, and spatial patterns of cell origins and lineages to determine the degree to which independently evolved but morphologically similar characters share the same developmental mechanisms. Extensive similarities would indicate strong constraints and fewer similarities would indicate more relaxed constraints.

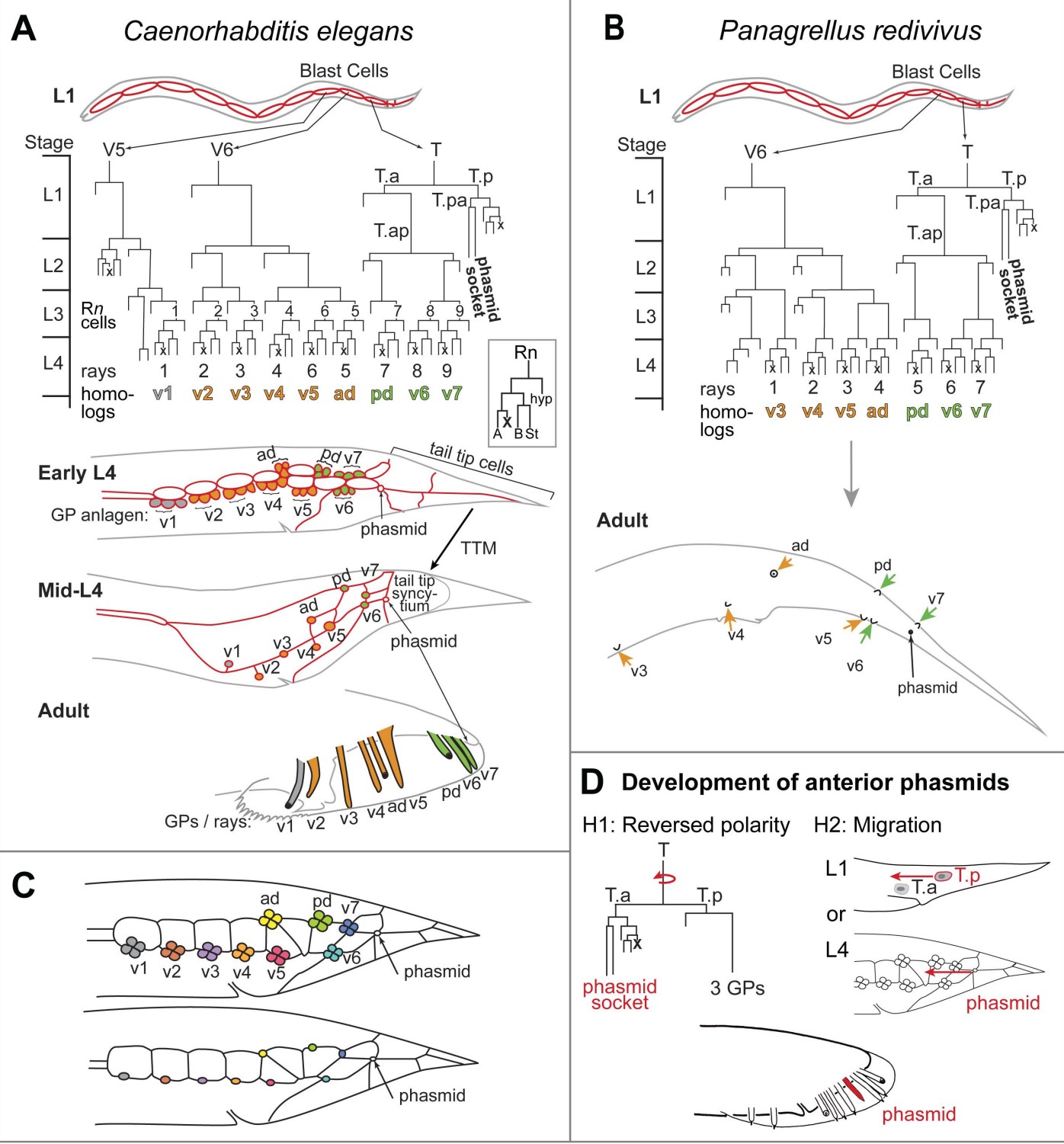

**Fig 1. Male tail characters. (A)** The *C. elegans* cell lineages of the three blast cells (V5, V6 and T) that make the rays (GPs) and phasmids. Three cells make each ray: one becomes the "structural cell" (st) and the other two become A- and B-type neurons (inset). After the anlagen for the GPs originate in the lateral hypodermis in L3, the ray cells cluster in early L4. Rays then differentiate, leaving the ray structural cells at the lateral surface (mid-L4) at about the same time as tail tip morphogenesis occurs [13]. After male tail morphogenesis is complete, the adult has finger-like rays in the

same anteroposterior order as the structural cells in the mid-L4 stage [11]. The rays were initially named 1-9, counting anterior to posterior in adult males. Names for homologous GPs are v1-v7, ad and pd [14]. Ray 5 (homologous to ad) is derived from a lineage posterior to that of ray 6. That is, R6 (= V6.ppppa) and R5 (= V6.ppppp) are labeled out of numerical order, with R6 being anterior instead of posterior to R5 in the cell lineage and relative positions of R*n* cells in the lateral epidermis. Originally, Sulston & Horvitz [15] had identified V6.ppppa as contributing to ray 6 and V6.ppppp to ray 5 in *C. elegans*. In their paper more fully characterizing *C. elegans* male development, Sulston et al. [16] labeled the R*n* cells in anteroposterior order (V6. ppppa as R5 and V6.ppppp as R6), but this was subsequently reversed (Fig 2 of [17]) to be consistent with the relative positions of the resulting rays in adult *C. elegans* males. For consistency, we follow the most recent (out-of-order) labeling of these R*n* cells and their descendants for all of the species characterized here. **(B)** The *Panagrellus redivivus* male cell lineages of the blast cells V6 and T**.** This species has three T cells: T1 on the left side and T2 + T3 on the right side. T1 has the lineage depicted here; T2 + T3 contribute to a similar lineage. V5 does not produce a GP; the most anterior lineage from V6 makes seam cells instead of a GP. Thus, adult males have 7 GPs and the homologs of v1 and v2 are absent in this species. **(C)** Schematic of the archetypical arrangement of GP cell groups and epidermal Rn.p cells early and late during GP development. The GP cell groups are color coded as in Figs 7–11). The archetype allows assignment of ray homologies across species [11,14]. **(D)** Two hypotheses for how phasmids in the anterior position could develop (see text for further details).

## Tail tip

We consider the shape of the tail tip because earlier phylogenetic analyses suggested that there were several changes between Pel and Lep tail forms [18]. Development of the male tail tip has been studied in great detail in *C. elegans*, which has short, round Pel tails in adults (Fig 1A) [13,19–27]. The Pel tail develops from a pointed larval tail through a process called Tail Tip Morphogenesis (TTM) during the last larval stage (L4) [13]. A similar process occurs in other species with Pel tails but not in species with Lep tails, which do not undergo TTM and retain the pointed larval shape into the adult stage, as in *P. redivivus* (Fig 1B) [21]. In *C. elegans,* TTM begins with apical fusion of the tail tip cells [13,22], but other species with Pel tails may perform TTM without fusions [21].

## Genital papillae

The numbers of GPs have changed recurrently in several lineages [12,28,29]; however, to determine if the same GPs have been lost or gained, GP homologies must first be identified. We know from the detailed cell-lineage analyses of *C. elegans* [16,17] that the GPs develop from nine bilateral pairs of R*n* neuroblast cells (R1-9 on both the left and right sides). A series of divisions of each R*n* cell produces one hypodermal cell (R*n*.p), two neurons (R*n*A and R*n*B) and one structural cell (R*n*st) (Fig 1A). The R*n* neuroblasts are derived from three blast cells V5, V6 and T. V5 generates one (R1), V6 generates five (R2–6) and T generates three (R7–9; Fig 1).

By immunostaining the adherens junctions (AJs) surrounding the epidermal cells in the tail, it was shown previously that the anlagen, which produce the GPs from the nine R*n* precursor cells, are arranged in a characteristic pattern where three of the R*n* anlagen—R5, R7 and R9—are dorsal of the others. The GP anlagen pattern is conserved in rhabditids and a generalized version was referred to as the archetype (Fig 1C) [11,30]. Because of this conservation, homologies can be assigned to each GP across different species, despite species-specific shifts in GP positions after their origin [11,28]. A naming system for GP homologs was adopted whereby the seven ventrolateral GPs on each side are designated "v1–v7," counting anterior to posterior, and the two subdorsal GPs produced by R5 and R7 are designated "ad" and "pd" respectively (for "anterior dorsal" and "posterior dorsal") [14]. Note that v7 is actually derived from the dorsal R9 anlage and in some species is positioned slightly dorsal to the v5 and v6 GPs. However, the "ad" and "pd" GPs (produced by R5 and R7, respectively) are prominently dorsal to the other GPs in all species characterized so far. The cells forming the "v" GPs do not shift ordinal position relative to each other (other than v7) but can migrate to different positions relative to the ad and pd GPs during L4, resulting in different anteroposterior orders of GPs in adults of different species [11,28].

This archetypal pattern of GP anlagen was established based on the investigation of a few species from different subclades within Rhabditina [28], but did not include representatives of the earliest branches of Rhabditina according to subsequent phylogenetic analyses [31], or the Diplogasteridae clade nested within Rhabditina. We were therefore interested in testing whether the archetypal GP pattern and the rules derived from it hold for other groups within and/or ancestral

to Rhabditina. To this end, we used AJ staining to investigate GP development in three additional Rhabditina species (*Cruznema tripartitum*, *Haematozoon subulatum*, *Poikilolaimus oxycercus*), a representative of Diplogastridae (*Diplogasteroides nasuensis*) and a representative of the outgroup (*Brevibucca saprophaga*).

The archetypal GP pattern includes 9 pairs of GPs. However, some species have only 8 pairs, for instance *Metarhabditis blumi*, for which AJ staining demonstrated that the R8 neuroblast does not divide and thus v6 is absent [11]. Here, we set out to determine which GP is missing in three other species with only 8 GPs, *C. tripartitum*, *P. oxycercus* and *B. saprophaga*.

## Phasmid position

In addition to the GPs, the rhabditid male tail bears one pair of phasmids. Using dyes selectively taken up by the chemoreceptive phasmids but not the mechanosensory GPs and scanning electron microscopy, previous work has shown that phasmids have changed position relative to GPs multiple times during evolution [11,12,30]. Specifically, whereas the phasmid is posterior to all GPs in some species (e.g., in *C. elegans*), it is anterior of the three posterior-most GPs in many other species [11,12]. For shorthand, we call these phasmid positions "posterior" versus "anterior." In both *C. elegans* and *P. redivivus*, the phasmid socket, which anchors the phasmid in the epidermis, and the three most posterior GPs on each side are derived from a T blast cell. The bilateral T blast cells (TL and TR) each produce the phasmid socket from their posterior daughter cell T.p and the posterior three GPs from their anterior daughter cell T.a (Fig 1A,B) [15–17]. In *C. elegans lin-44/Wnt* loss-of-function mutations, this polarity is reversed, such that phasmid socket-like lineages are produced from T.a and GP-like lineages from T.p [32]. Thus, one hypothesis (H1, Fig 1D) for a developmental event switching phasmid position from posterior to anterior is a polarity reversal of the first T cell division, such as might be produced by a heterochronic delay in the expression of the LIN-44/Wnt signaling ligand or its receptor relative to its expression in *C. elegans* [12,28]. An alternative hypothesis (H2, Fig 1D) is that no such reversal occurs and instead the cells producing the phasmid socket migrate anteriorly from an originally posterior position. In the work reported here, we tested hypotheses regarding cell origins, cell migrations, and other developmental events underlying these male tail characters.

## Results

### Phylogenetic analysis

Parsimony was used to trace character evolution on a robust cladogram (Fig 2) previously inferred from molecular data [31] and amended based on data from other studies [33–35]. It should be noted that a recent phylogenetic analysis with data from whole genomes resulted in a different branching order for one important taxon, Diplogastridae [36]. That analysis placed Diplogastridae as the second most anciently diverged branch of Rhabditina after *Poikilolaimus*, whereas Kiontke et al. [31] found that Diplogastridae + *Rhabditoides* form the third most ancient branch after the Pleiorhabditis group + *Haematozoon*. All other branching orders are in agreement between the two analyses. Because the whole genome [36] analysis included fewer phylogenetically representative rhabditid taxa and in particular did not include *Rhabditoides* and *Haematozoon*—taxa that help to determine the lower branching orders in the clade and are critical for interpreting character evolution [31]—we decided to use the more taxon-rich phylogeny [31] as the framework for the current analysis. An additional justification for this is that *Rhabditoides* is the sister taxon to Diplogasteridae, yet diverged after Pleiorhabditis. Thus, until whole-genome analyses include such taxa, we have good reason to believe that our phylogeny is robust. However, we also note the single instance (phasmid position) in which these different phylogenetic positions for Diplogastridae affect our interpretation of male tail character evolution.

One last thing to note about the phylogeny used here [31] is the lack of resolution in the branching order of Bunonematidae and Brevibuccidae. However, in other molecular phylogenies and nematode taxonomic systems, this is resolved with Bunonematidae as the first-diverging branch within Rhabditina and Brevibuccidae as a sister group to Rhabditina [30].

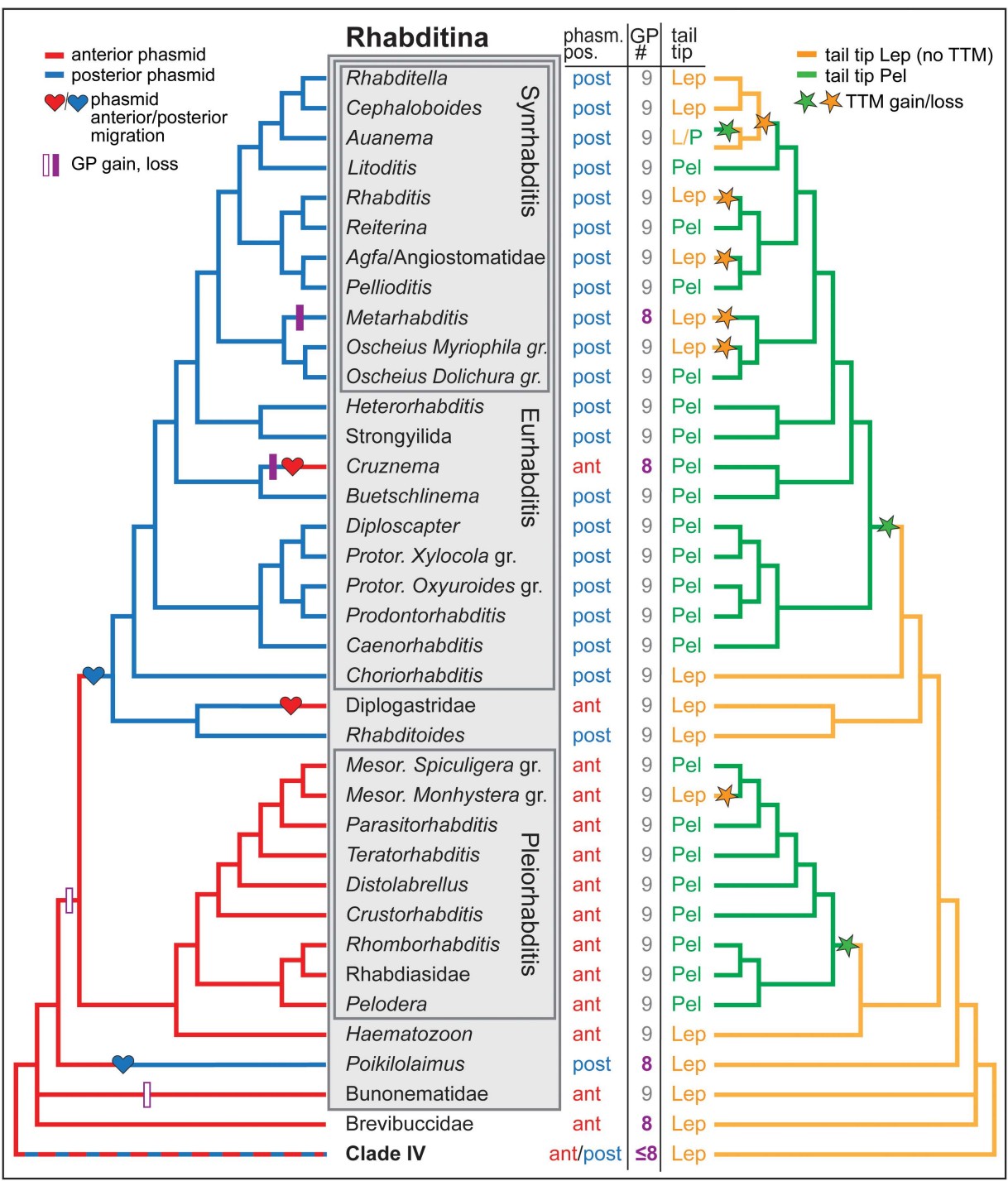

**Fig 2. Male tail character evolution on the phylogeny of Rhabditina.** The phylogeny is based on molecular characters [31], with taxa designated as in [34]. See text for more information about the phylogeny and character tracing.

Recurrent evolution was found for the three characters of interest here ([Fig 2](Fig 2)):

(1) The shape of the tail tip changed 8 times. A leptoderan tail tip is the ancestral condition in Rhabditina. Tail tip morphogenesis (TTM) was unequivocally gained three times to produce a peloderan (Pel) tail tip, once within *Auanema,* once in the stem species of Pleiorhabditis, and once in the stem species of Eurhabditis. There is another appearance of Pel tails that was not investigated here: From within diplogastrids with Lep tails, *Goodeyus* males evolved Pel tails. Loss of TTM occurred unequivocally in the lineage to the *Monhystera* species group of *Mesorhabditis*. Recurrent evolution involving TTM also occurred in other taxa of the Synrhabditis group, but the polarity of these changes is equivocal (assuming losses are equally likely as gains), either one loss plus four gains or 5 losses. Here, we propose to resolve this ambiguity using the Dollo-like assumption that gains of this complex character are less likely than losses. We feel this is well justified because loss of morphogenesis could potentially be due to any of a number of simple loss-of-function mutations, whereas gain likely involves the specific co-option of a complex male-specific morphogenetic program that involves the co-ordinated regulation of many cytological processes [19–27]. Under this assumption (equivalent to applying DELTRAN, delayed transformation [37]), the 5 losses of TTM occurred in the stem species of individual lineages within this group: the ancestral lineage to *Rhabditella + Cephaloboides + Auanema*, *Rhabditis*, *Agfa +* Angiostomatidae, *Metarhabditis*, and the *Oscheius Myriophilus* species group. This conclusion is unaffected by the different proposed placements of Diplogastridae.

(2) There were four recurrent changes in the number of GPs. First, it is unequivocal that one GP pair was lost in *Metarhabditis* and in *Cruznema*. If Brevibuccidae is sister to Rhabditina (including Bunonematidae) [30], then there are two equally parsimonious scenarios for the other two changes: (a) The Rhabditina stem species had 8 bilateral pairs of GPs like Brevibuccidae and several species in the outgroup (others have fewer GPs), and one GP pair was gained both in Bunonematidae and in the rhabditid stem lineage after the divergence of *Poikilolaimus*. In this case, the 8 GP pairs in *Poikilolaimus* is a plesiomorphic (ancestral) character state. (b) Alternatively, the stem species of Rhabditina gained one pair of GPs and had 9 GP pairs. In this case, the 8 GP pairs in *Poikilolaimus* is derived (i.e., a reversal due to loss of a GP pair).

(3) Four recurrent changes happened in the position of the phasmid relative to the GPs. The phasmid was in the anterior position in the stem species of Rhabditina and switched twice to the posterior position, in *Poikilolaimus* and after the divergence of Pleiorhabditis and *Haematozoon*. Reversals to anterior phasmids then occurred in the stem species of diplogastrids and *Cruznema*. If Diplogastridae actually diverged before Pleiorhabditis + *Haematozoon*, then the anterior phasmid position in diplogastrids would be plesiomorphic instead of a derived feature, reducing the number of recurrent changes to three instead of four.

## Laser ablations

To determine which daughter of the T cell produces the phasmid socket in species in which it is located anterior of the 3 posterior GPs in adult males, we laser-ablated cells in the T lineage and observed the resulting phenotype in adults. We performed these experiments in three species with anterior phasmids, *P. strongyloides, C. tripartitum* and *D. nasuensis,* and in one species with posterior phasmids, *P. oxycercus.*

To identify the T cells or their daughters, we used the detailed information for the *C. elegans* cell lineage [15–17] as a guide. The cell lineage of *P. redivivus* is nearly identical to that of *C. elegans* ([Fig 1](Fig 1)) [38]. All of the species we investigated are more closely related to *C. elegans* than to *P. redivivus*. We therefore assumed that nuclei found in similar positions and numbers and with similar appearance as in both *C. elegans* and *P. redivivus* are homologous in all species. Using this logic, we identified epidermal cells by the "fried-egg"-appearance of their nuclei and distinguished these from neuronal cell nuclei, which have a stippled appearance with no distinguishable nucleolus (Sulston & Horvitz 1976). We also observed

five hypodermal nuclei in the tail tip that we homologized with the nuclei of the *C. elegans* tail tip cells hyp8–11. One or two additional nuclei, located on the ventral side near the anus, were identified as hyp13 in males and hyp7 in females. We could also locate the B cell nucleus by its position posterior to the anus. Using these cells as landmarks, we looked for large hypodermal nuclei on the lateral surface to find the T cells or their progeny for ablations (Table 1).

**Pelodera strongyloides.** In *P. strongyloides* strain DF5022, we first observed the T cell division in a few young L1 larvae and found that it happens in an a-p (anteroposterior) fashion as expected (Fig 3, S1–S5 Figs). This allowed us to unambiguously identify the T daughters in the tail of other animals. There was no evidence of migration of the T-cell daughters before their next division. We then performed ablations (Table 1). Ablation of the T cell in females led to the absence of the phasmid (n = 4) on the operated body side. When T was ablated in males, the phasmid and the three posterior rays on the respective body side were missing (n = 2). This confirms that the cell we identified as T is the progenitor of three GPs and the phasmid socket cells as in *C. elegans* and *P. redivivus* [38]. In females, only ablation of T.p led to missing phasmids in the adult (n = 1), ablation of T.a had no effect (n = 8). When T.a was ablated in males (n = 7), the three posterior GPs were absent on the operated body side; when T.p was ablated, the phasmid was missing (n = 6). Thus, in *P. strongyloides*, the phasmid socket is the product of the posterior daughter of T and the three posterior GPs of males are descendants of T.a, as in *C. elegans* and *P. redivivus*.

**Cruznema tripartitum.** Using the same criteria as in *P. strongyloides,* we unambiguously identified T and its daughters in *C. tripartitum* strain SB202 (Fig 4, S6–S11 Figs). Ablation results were similar to those in *P. strongyloides* (Table 1): Ablation of T and T.p in females resulted in the absence of the phasmids (n = 4). We also ablated T.pa in females, which also resulted in the absence of the phasmid (n = 4). In males, ablation of T led to absence of the phasmid and of only two GPs (n = 2). Ablation of T.p resulted in the absence of the phasmid (n = 4). Ablations of T.a did not succeed, but ablations of T.ap led to the absence of two GPs on the respective body side (n = 4).

**Diplogasteroides nasuensis.** Like all diplogastrids, *D. nasuensis* (strain SB335) undergoes the first larval molt inside the eggshell and larvae hatch as second juveniles. By that time, the T lineage has undergone two divisions such that four granddaughter cells are present (Fig 5, S12–S14 Figs). Two of the nuclei of these cells are located further dorsal than the other two. Assuming that the division pattern of the T daughters is the same as in *C. elegans* and that no cell migrations happen in the meantime, we infer the identity of the nuclei as shown in Fig 5A. The phasmid opening is located just apical of the nucleus of the cell identified as T.pa. We ablated one, two or three of the nuclei and observed the effect on adult morphology (Table 1). Given our identification of the cells in L2, ablation of the T.p progeny led to the absence of the phasmid in females (n = 8) and males (n = 2), and ablation of the T.a progeny in males led to the absence of 3 posterior rays (n = 2).

In *D. nasuensis*, we also ablated the large nucleus of another laterally positioned blast cell, which we assumed to be V6. Ablation of this cell in males led to the absence of 5 GPs in adults, including the most anterior GP (n = 4). Thus, V6 produces 5 GPs as in *C. elegans.* Four GPs remain: the three T GPs and one GP near the cloaca and slightly dorsally located. This GP must be v1, the GP derived from V5. On the non-ablated side the corresponding GP is positioned near the second ventral GP but further dorsal. v1 thus constitutes the third (most anterior) *dorsal* GP characteristic of diplogastrids.

**Poikilolaimus oxycercus.** We attempted ablation of T and its daughters in 15 L1s of *P. oxycercus* strain EUK103 (Fig 6, S15, S16 Figs). We obtained only two adults with an altered phenotype: one female with a missing phasmid upon ablation of the T cell, and one male with three posterior GPs missing after ablation of T.a (Fig 6B). This result shows that three posterior GPs are derived from the T.a cell, and we infer that the phasmid socket is derived from T.p.

## MH27 staining

We stained the adherens junctions with the MH27 antibody in 4 species and took z-stack-photographs of the tails of L3 and L4 males during GP development. The goal was to obtain a series of image stacks for different developmental steps from the time when the R*n* blast cells are not yet divided until the fully developed GPs are fixed at their final positions.

**Table 1. Cell ablation experiments[1].**

| Species | Individual | Cell(s) ablated | Sex | Result: feature(s) missing |
|---|---|---|---|---|
| *P. strongyloides* | 12 | T l | female | phasmid l |
| | 17 | T l | female | phasmid l |
| | 4 | T r | female | phasmid r |
| | 14 | T r | female | phasmid r |
| | 26 | T.a l | female | nothing |
| | 7 | T.a r | female | nothing |
| | 10 | T.a r | female | nothing |
| | 19 | T.a r | female | nothing |
| | 21 | T.a r | female | nothing |
| | 23 | T.a r | female | nothing |
| | 37 | T.a r | female | nothing |
| | 25 | T.a. l | female | nothing |
| | 27 | T.p r | female | phasmid r |
| | 16 | T r | male | phasmid and 3 post. GPs r |
| | 44 | T r | male | phasmid and 3 post. GPs r |
| | 40 | Tl | male | phasmid and 3 post. GPs l |
| | 20 | T.a l | male | 3 post. GPs l |
| | 34 | T.a l | male | 3 post. GPs l |
| | 42 | T.a l | male | 3 post. GPs l |
| | 46 | T.a l | male | 3 post. GPs l |
| | 47 | T.a l | male | 3 post. GPs l |
| | 41 | T.a r | male | 3 post. GPs r |
| | 33 | T.a r | male | 3 post. GPs r |
| | 18 | T.p l | male | phasmid l |
| | 36 | T.p l | male | phasmid l |
| | 45 | T.p l | male | phasmid l |
| | 13 | T.p r | male | phasmid r |
| | 28 | T.p r | male | phasmid r |
| | 31 | T.p r | male | phasmid r |
| | | | | |
| *C. tripartitum* | 9 | T r | female | phasmid r |
| | 21 | T. l | female | phasmid l |
| | 30 | T.p l | female | phasmid l |
| | 43 | T.p r | female | phasmid r |
| | 40 | T.pa l | female | phasmid l |
| | 27 | T.pa r | female | phasmid r |
| | 35 | T.pa r | female | phasmid r |
| | 32 | T.pa + T.ap r | female | phasmid r |
| | 17 | T r | male | phasmid and 2 post. GPs r |
| | 19 | T. l | male | phasmid and 2 post. GPs l |
| | 15 | T.p r | male | phasmid r |
| | 16 | T.p l | male | phasmid l |
| | 26 | T.p r | male | phasmid r |
| | 31 | T.p l | male | phasmid l |
| | 37 | T.ap l | male | 2 post. GPs l |

*(Continued)*

**Table 1.** (Continued)

| Species | Individual | Cell(s) ablated | Sex | Result: feature(s) missing |
|---|---|---|---|---|
| | 38 | T.ap l | male | 2 post. GPs l |
| | 25 | T.ap l | male | 2 post. GPs l |
| | 28 | T.ap r | male | 2 post. GPs r |
| | | | | |
| *D. nasuensis* | 24 | T.ap l | female | nothing |
| | 51 | T.ap l | female | nothing |
| | 9 | T.ap l | female | nothing |
| | 21 | T.ap r | female | nothing |
| | 44 | T.ap r | female | nothing |
| | 1 | T.ap, T.pa, T.pp l | female | phasmids l |
| | 48 | T.pa l | female | phasmid l |
| | 45 | T.pa r | female | phasmid r |
| | 46 | T.pa r | female | phasmid r |
| | 37 | T.pa r | female | phasmid r |
| | 20 | T.pa, T.pp l | female | phasmid l |
| | 31 | T.pa, T.pp l | female | phasmid l |
| | 38 | T.pa, T.pp r | female | phasmid r |
| | 17 | T.pp l | female | nothing |
| | 34 | T.pp l | female | nothing |
| | 11 | T.aa l | female | nothing |
| | 6 | V6 l | female | nothing |
| | 7 | V6 l | female | nothing |
| | 19 | V6 l | female | nothing |
| | 18 | T.ap l | male | 3 post. GPs l |
| | 35 | T.ap l | male | 3 post. GPs l |
| | 40 | T.ap, T.pa, T.pp l | male | phasmid + 3 post. GPs l |
| | 50 | T.pa l | male | phasmid l |
| | 26 | T.pa r | male | phasmid r |
| | 15 | T.pp l | male | nothing |
| | 29 | T.pp l | male | nothing |
| | 23 | T.pp r | male | nothing |
| | 27 | T.pp r | male | nothing |
| | 5 | V6 l | male | 5 GPs l |
| | 36 | V6 l | male | 5 GPs l |
| | 42 | V6 r | male | 5 GPs r |
| | 47 | V6 r | male | 5 GPs r |

[1]Results of ablation for each individual animal (designated by species name and a unique number). The designated cell or cells was/were ablated on one side of the animal (l = left side, r = right side). Adult morphological features/organs that were affected are then noted.

**Poikilolaimus oxycercus.** Adults of *P. oxycercus* have 8 pairs of GPs on each side and posterior phasmids (Fig 7, S17–S20 Figs). Early during GP development, 9 R*n* blast cells are present on each side and the phasmids are positioned posterior to all of them. Later, eight GP cell groups per side are discernible. Between the first and second is a large undivided cell, inferred to be the undivided R2 neuroblast. Three GP cell groups (ad, pd and v7) are positioned further

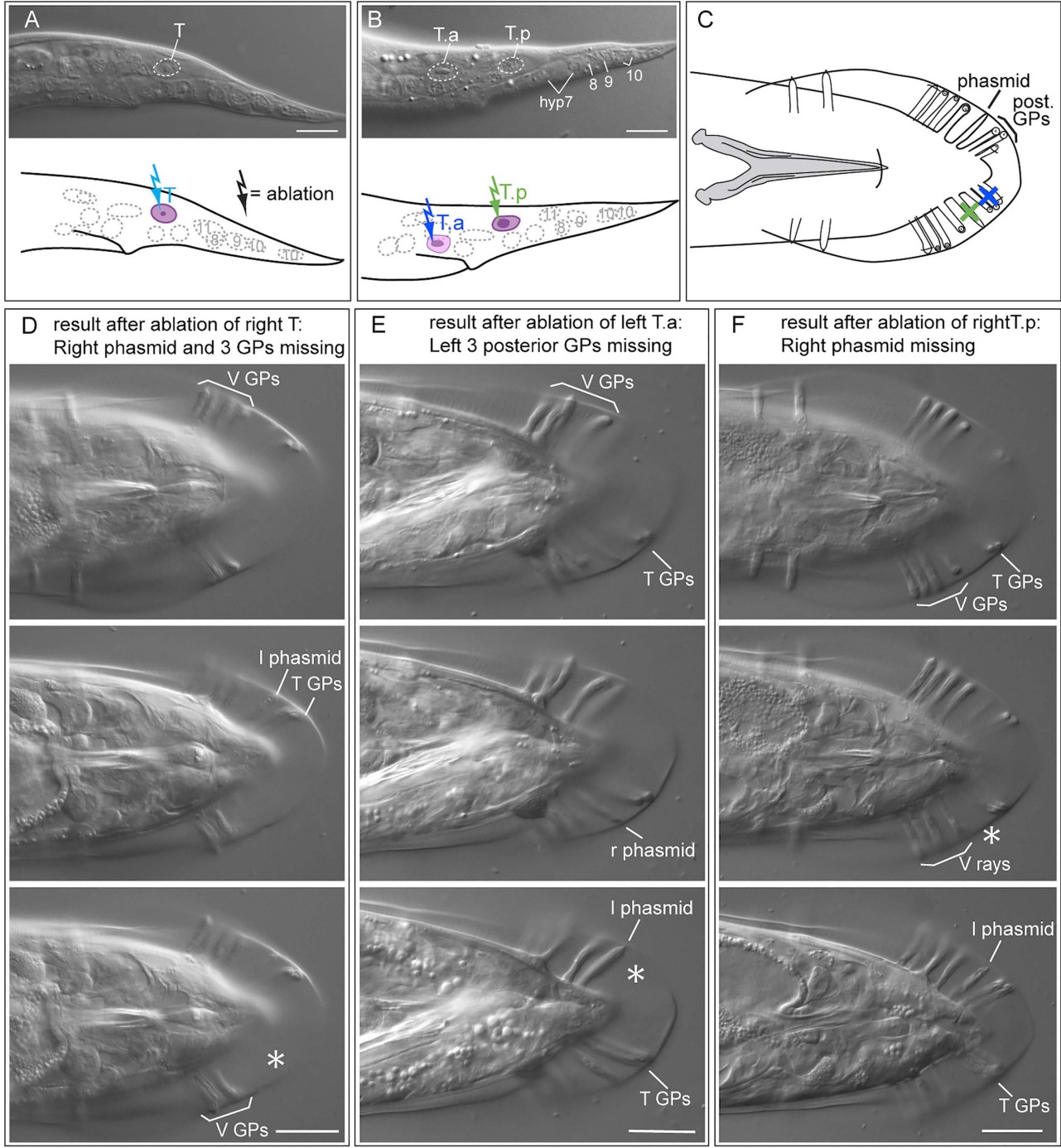

**Fig 3. Results of ablations of T lineage cells in *Pelodera strongyloides* DF5022. (A)** DIC image (top) and schematic drawing (bottom) of the tail of an early L1 larva before the first division of T. Some nuclei are outlined. Numbers in the nuclei refer to the hyp cells in the tail tip. **(B)** DIC image (top) and schematic drawing (bottom) of the tail of an L1 larva after the first division of T. Colored arrows indicate which cells were ablated, scale bars in A and B = 10 μm. **(C)** Drawing of the tail of an adult male in ventral view. A blue X indicates the GPs missing upon ablation of T.a, a green X indicates the phasmid that is missing upon ablation of T.p. **(D, E, F)** Examples of the result upon ablation of T or its progeny. DIC images of three focal planes taken in ventral view are shown for each animal. Scale bars = 20 μm. **(D)** Male #44 (see Table 1): after ablation of the right T cell, the phasmid and the three posterior (T) GPs are missing on the right side (asterisk); the left phasmid and T GPs are present and pd is out of focus. **(E)** Male #46: after ablation of the left T.a; the left three T GPs are missing and phasmids are present on both sides. **(F)** Male #13: after ablation of the T.p; the left phasmid is missing and all T GPs are present. Raw image data are provided in S1–S5 Figs.

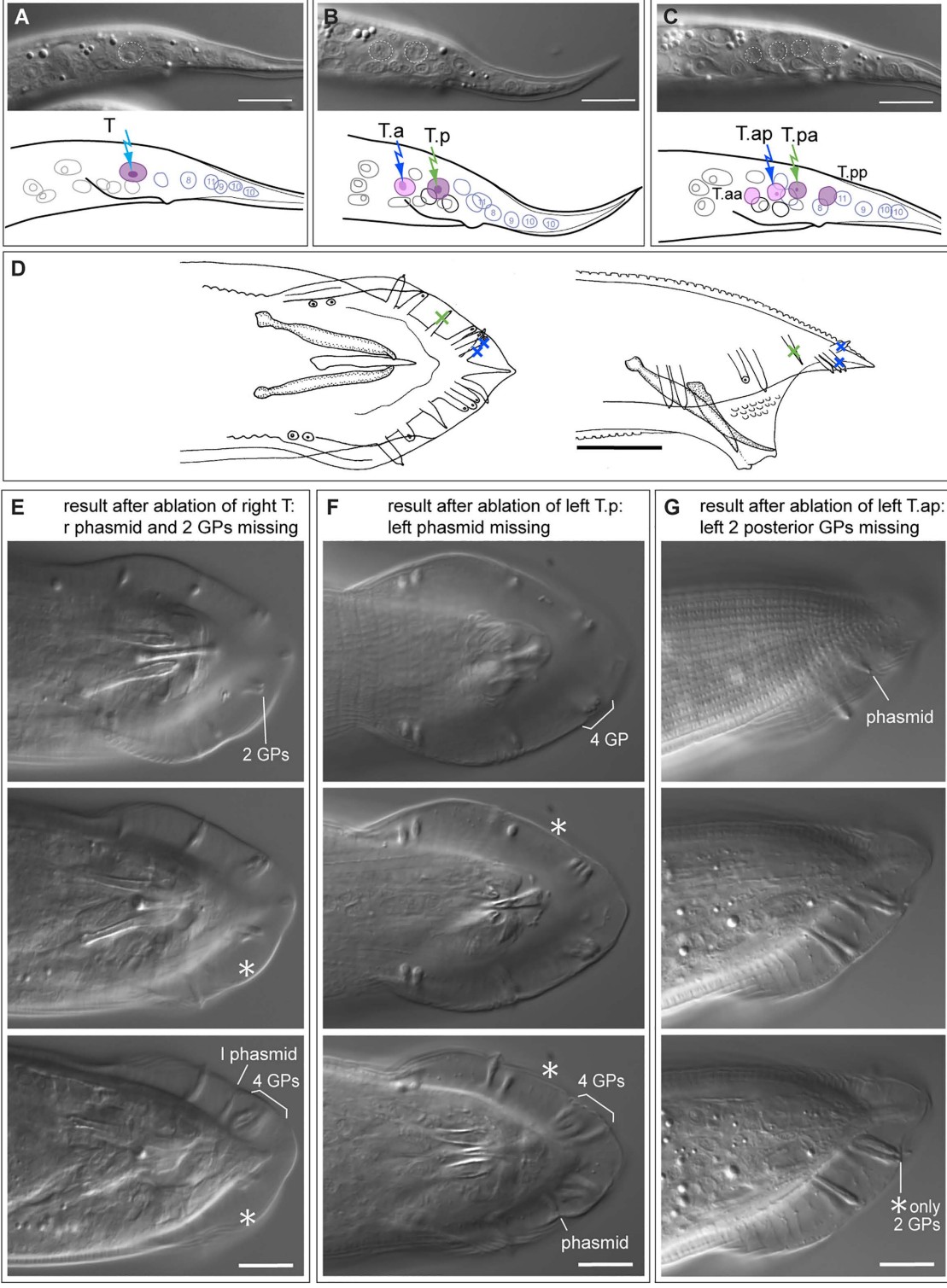

**Fig 4. Results of ablations of cells in the T lineage in *Cruznema tripartitum* SB202. (A)** DIC image (top) and schematic drawing (bottom) of the tail of an early L1 larva before the first division of T. Some nuclei are outlined. Numbers refer to the hyp cells in the tail tip. **(B)** DIC image (top) and schematic drawing (bottom) of the tail of an L1 larva after the first division of T. **(C)** DIC image and schematic drawing of DIC image (top) and schematic drawing (bottom) of the tail of an L1 larva after the second division of T. Colored arrows indicate which cells were ablated, scale bars in **A**, **B**, **C** = 10

μm. **(D)** Drawings of the tail of an adult male in ventral (left) and lateral (right) view. A blue X indicates the GPs missing upon ablation of T.a and a green X indicates the phasmid that is missing upon ablation of T.p; scale bar 20 μm. **(E, F, G)** Examples of the result upon ablation of T or its progeny. DIC images of three focal planes are shown for each animal. Scale bars = 20 μm. **(E)** Male #17 (ventral view): after ablation of the right T cell, the phasmid and two of the four posterior GPs are missing on the right side (asterisk). **(F)** Male #31 (ventral view): after ablation of the left T.p, the left phasmid is missing (asterisk). **(G)** Male #25 (left side view): after ablation of the left T.ap, two of the four posterior GPs are missing on the left side (asterisk). Raw image data are provided in S6–S11 Figs.

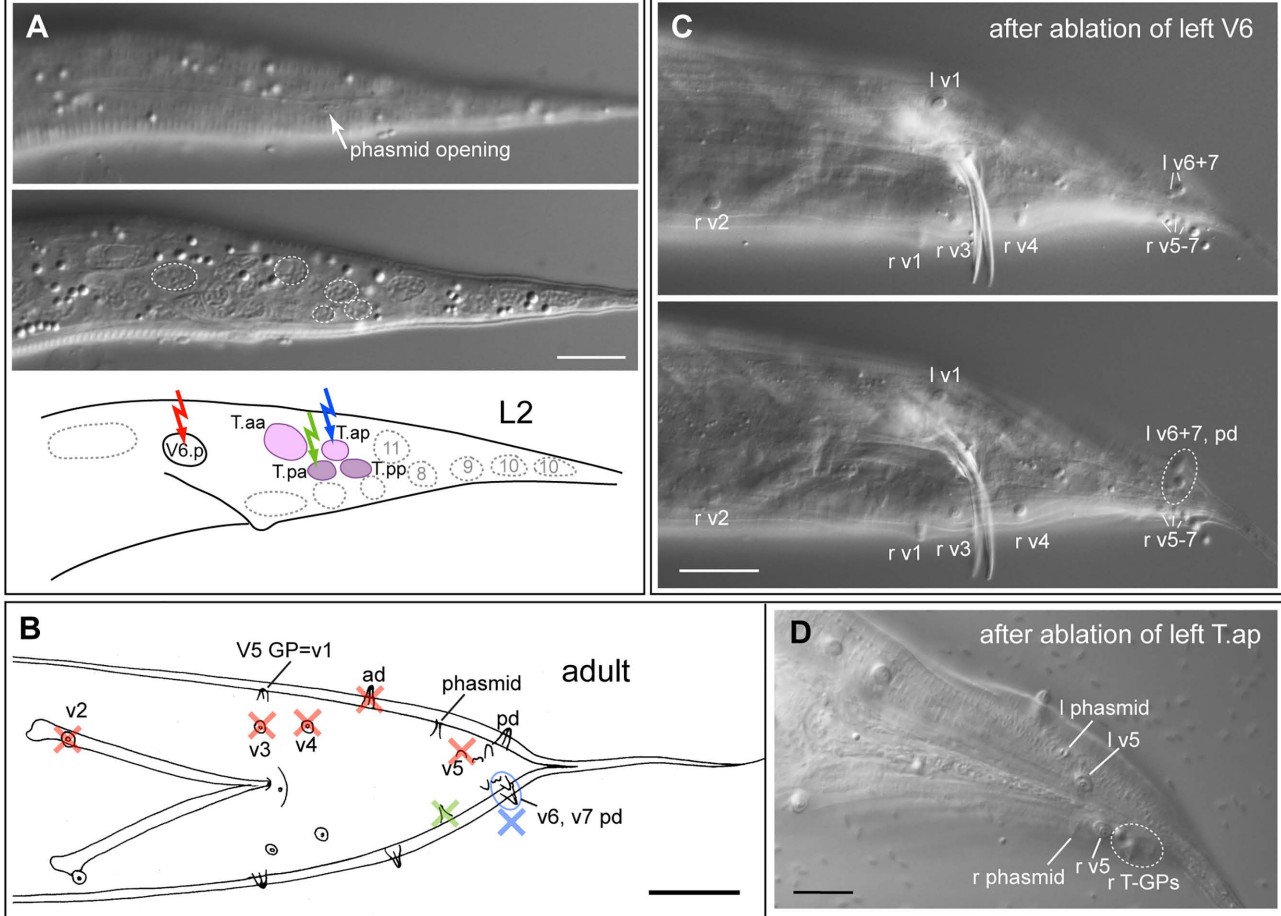

**Fig 5. Results of ablations of cells in the T lineage in *Diplogasteroides nasuensis* SB335. (A)** DIC images (top) and schematic drawing (bottom) of the tail of an early L2 larva right after hatching. The top DIC image is focused on the apical surface and shows the location of the phasmid opening. The identity of the T-lineage cells as marked in the schematic was derived from the ablation results. Colored arrows indicate which cells were ablated, scale bar = 10 μm. **(B)** Schematic drawing of the tail of an adult male in ventral view. Red Xs indicate GPs that are missing when V5.p is ablated, a blue X marks GPs that were missing when the cell identified as T.ap is ablated, and a green X marks the phasmid that is missing when T.pa is ablated. **(C)** Male #36 (tail in ventral view): after ablation of the left V6.p cell, five GPs are missing. The remaining rays are v1 near the cloaca and three in the posterior (v6, v7 and pd). **(D)** Male #18 (posterior end of the tail in ventral view): after ablation of the left T.ap, three GPs (v6, v7, pd) are missing but the left phasmid is present. Raw image data are provided in S12–S14 Figs.

dorsal. By the time GP development is almost complete, the hypodermal R3.p-R9.p have fused to form two large cells. R1.p and the undivided R2 are not fused with this syncytium. The surface area of these epidermal cells enlarges during L4 development and they are unusually broad in this species. By stage 4, the GP cell groups v1, v3 and v4 separate from their epidermal sister cell and migrate slightly ventrally.

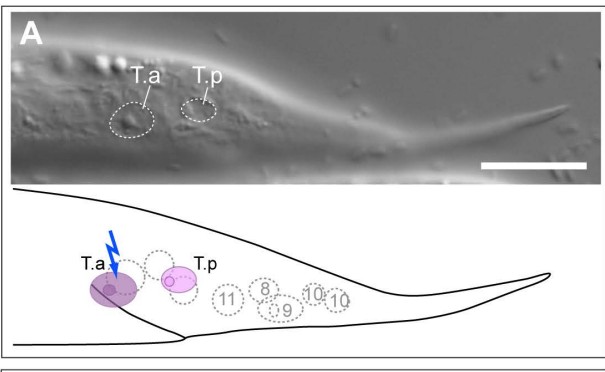

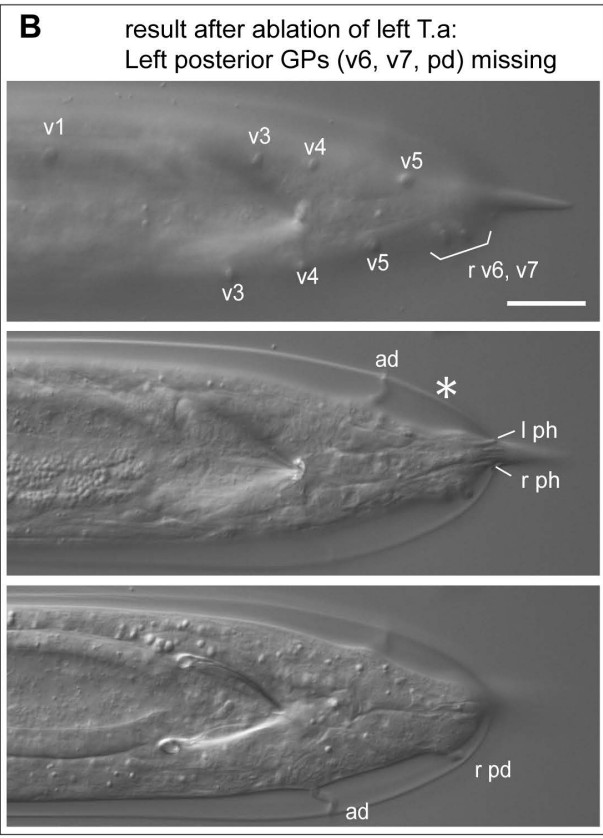

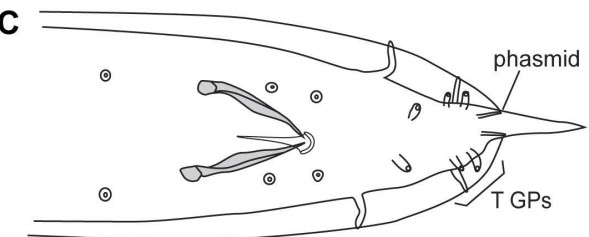

**Fig 6. Results of ablations of cells in the T lineage in *Poikilolaimus oxycercus* EUK103. (A)** DIC image (top) and schematic drawing (bottom) of the tail of an L1 larva after the first division of T. **(B)** Male after ablation of the left T.a: the left posterior GPs are missing (asterisk). **(C)** Schematic drawing of a *P. oxycercus* male in ventral view. Scale bar in **A** = 10 μm, in **B** = 20 μm. Raw image data are provided in S15, S16 Figs.

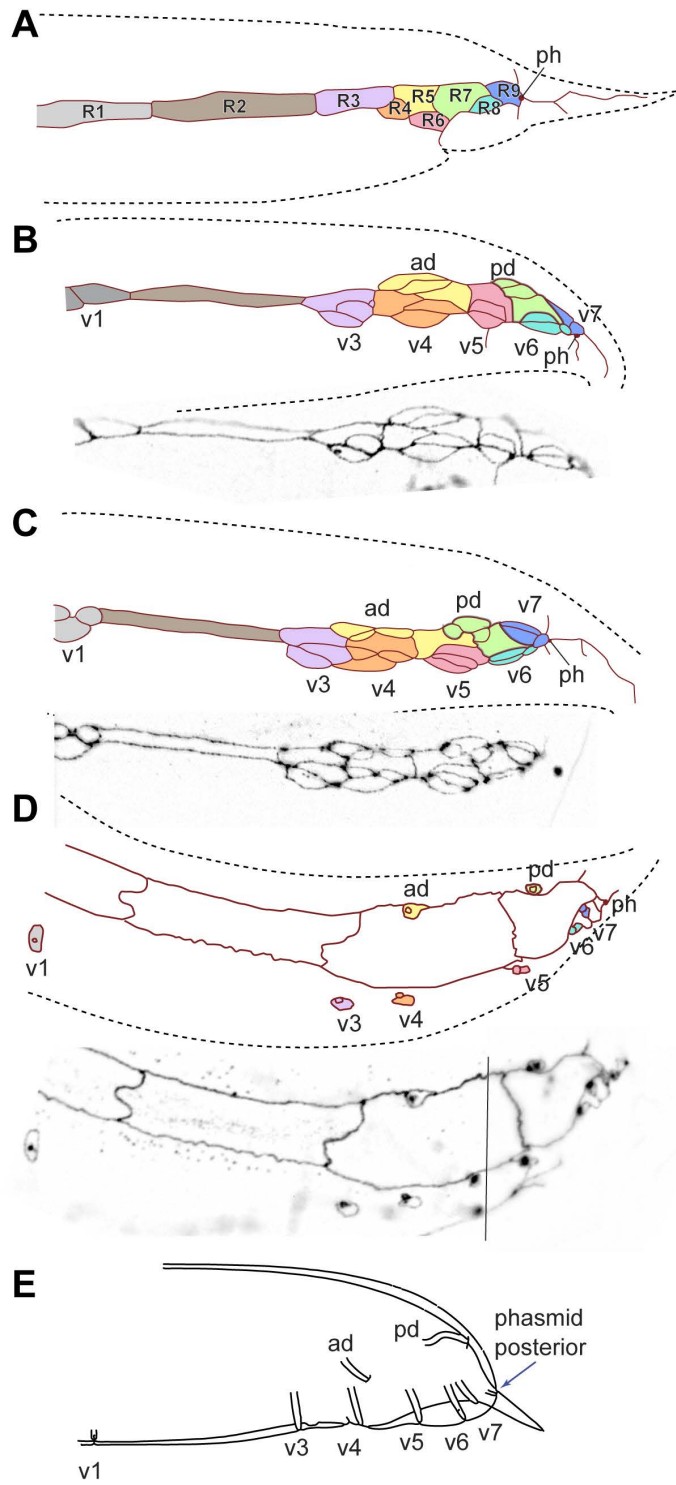

**Fig 7. MH27 antibody staining of adherens junctions in thetail of male *P. oxycercus* (strain EUK106) larvae.** Reconstructions of the adherens junction patterns (top) and part of the z-projected photomicrographs it is based on (bottom) for **(A, B)** two early L3 males, **(C)** a stage 1 male and **(D)** a stage 3 male (stages as in [11]). **(E)** Schematic drawing of an adult male. In **(D)**, the photomicrographs are from two different z-plane merges separated by a line. Cells and GP cell groups are labeled based on our hypotheses for the homologies of these structures, and cells from the same lineage are marked with the same color. *P. oxycercus* has only 8 GP cell groups. In **(B–D)**, a large undivided cell (brown) that we think is R2 separates the most anterior GP group (v1) from the next (v3). Raw image data are provided in S17–S20 Figs.

***Haematozoon subulatum.*** *H. subulatum* (strain SB303) has 9 pairs of GPs and phasmids anterior of 4 GPs on each side. Early during GP development, 9 R*n* blast cells could be identified (Fig 8, S21–S24 Figs). R5 and R7 touch and R6 is displaced ventrally. The phasmid is located at the border between R6 and R7. Later, three GP cell groups are located dorsally, which we homologize with ad, pd and v7. The v5 cell group shifts posterior of the phasmid. The GP cell groups v1-v4 move ventrally and away from the R1-4.p cells early before the third division of the GP lineage. By stage 4, R1-4.p have fused with the body seam. The final fate of the other Rn.p cells is not known.

***Diplogasteroides nasuensis.*** *D. nasuensis* (Fig 9, S25–S28 Figs) has 9 pairs of GPs of which three are positioned dorsally. On each side, the phasmid is located anterior of 4 GPs. At the earliest observed stage, only R3-R9 could be clearly identified. These cells are rounder than the seam cells that are further anterior. R3-R9 are mostly arranged in a single row, but one cell is ventral to the others. This cell is the precursor of v5 as can be seen in subsequent stages. The precursors of ad and pd touch, and the precursor of ad touches R4. Early during development, the v4 cell group was easily identified. Further anterior, however, there were initially only two groups of dividing cells (Fig 9B). Later, it appears as if anterior of the v4 group, two cell groups are positioned dorsoventrally relative to each other; another group is further anterior (Fig 9C). Which of these are v1, v2 and v3 was unclear from the staining patterns alone, but the ablations described above showed that the dorsal GP is the v1 homolog. Staining of males during an early stage of GP development showed that the phasmid is located anterior of R7. During divisions of the R*n* neuroblasts, the phasmid is temporarily located slightly posterior of the v6 cell group, but by stage 4, the v5 cell group is seen posterior of the phasmid as in adults.

***Brevibucca saprophaga.*** *B. saprophaga* strain SB261 (Fig 10, S29–S31 Figs) has 8 pairs of GPs, and the phasmid is located anterior of 3 GPs, v5, v6 and v7. For this species, we could not obtain stainings of a male early during GP development. The arrangement of cells at the earliest stage available suggests the following: There are eight GP cell groups; three GP cell groups are located dorsally, identifiable as ad, pd and v7; the precursor of the anteriormost three GPs separate from their Rn.p cell and migrate ventrally very early. Between the two anteriormost Rn.p cells lies a hypodermal cell which we interpret as the undivided R2. The phasmid is initially located posterior of the v7 cell group, but by stage 4, it is located anterior of the v5, v6 and v7 cell groups, as in adults.

## Discussion

Our phylogenetic analysis indicated that all three characters are homoplastic: Tail tip shape changed 8 times, GPs were gained and lost twice and the phasmid changed its position 4 times. We can thus use our analysis of the development of these characters to find evidence of constraints.

### Tail tip

The development of the tail tip in several Rhabditina species has been studied previously [18,21,28]. It was shown that a short Pel tail can develop via TTM with apical fusion of the tail tip cells in *C. elegans* and without fusion in all other studied Pel species, *P. strongyloides, Teratorhabditis palmarum* and *Reiterina typica*. In at least one species with a Lep tail tip, *M. blumi,* some of the tail tip cells appear to fuse, even though TTM does not happen. This species belongs to a lineage in which TTM was lost and the Lep tail evolved secondarily. In other Lep species, the tail tip cells remain unretracted and unfused. Here, we studied the AJs in additional species with Lep tails, but we did not follow their development to adult-hood. We therefore do not know if fusions of the tail tip cells occur late during development in these species and did not reach any different conclusions than previously, i.e., that there may not be a constraint requiring cell fusion to co-occur with TTM [28]. Ongoing studies are comparing gene expression profiles of tail tips in different species with the goal of understanding the extent of constraint versus evolvability in the gene regulatory network underlying tail tip morphogenesis.

### GP homology

In order to trace the evolution of GP number and phasmid position, we first needed to determine homologies among the GPs. Previous homology hypotheses have been derived from the position of the GPs in adult males in conjunction

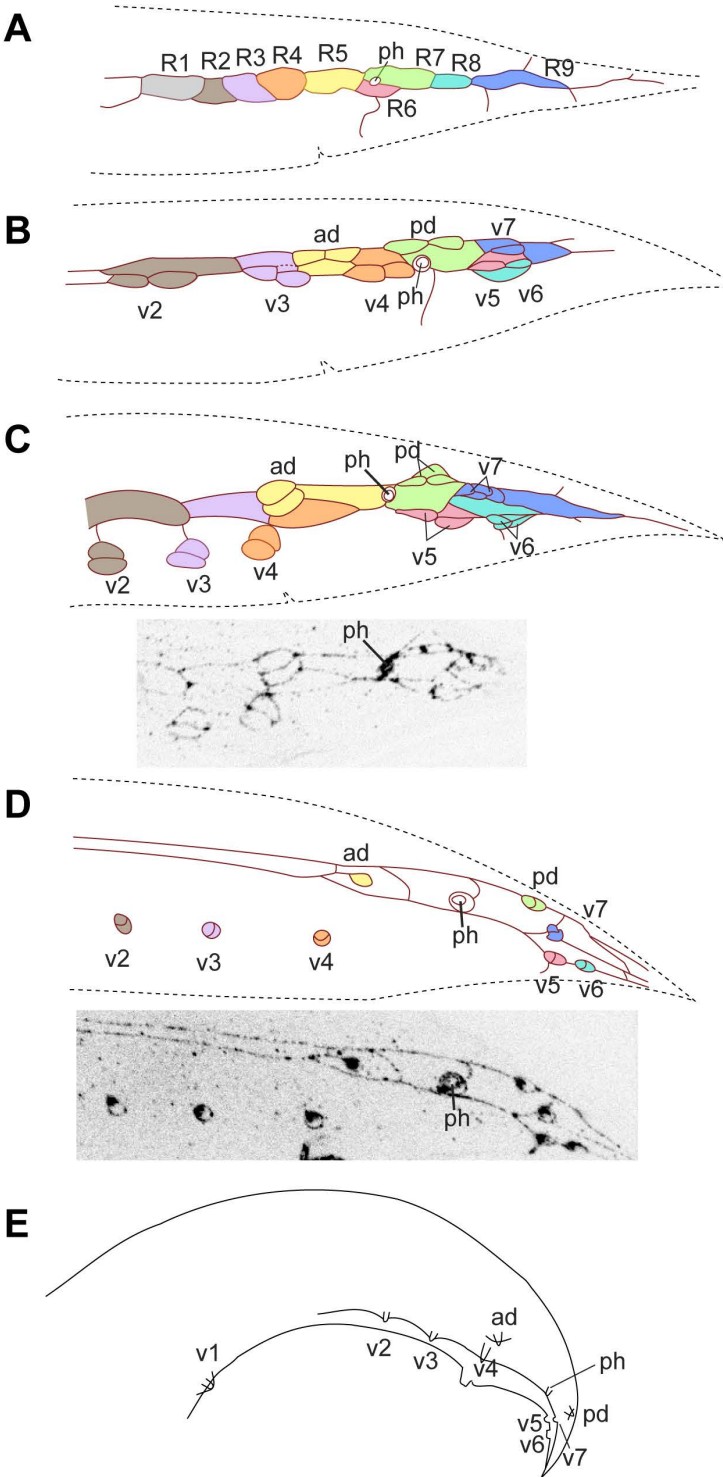

**Fig 8. MH27 antibody staining of adherens junctions in the tail of male *H. subulatum* SB303 larvae.** Reconstructions of the adherens junction patterns of **(A, B)** males early during GP development, **(C)** a stage 1 male and **(D)** a stage 3 male (stages as in [11]). **(E)** Schematic drawing of an adult male. In **(C and D)**, part of the z-projected photographs is shown that are the basis for the reconstruction. The most anterior GP cell group (V1) was outside of the captured images in **(B–D)**. In this species, the phasmid opening is positioned anterior of GP groups v5–7 throughout GP development. Raw image data are provided in S21–S24 Figs.

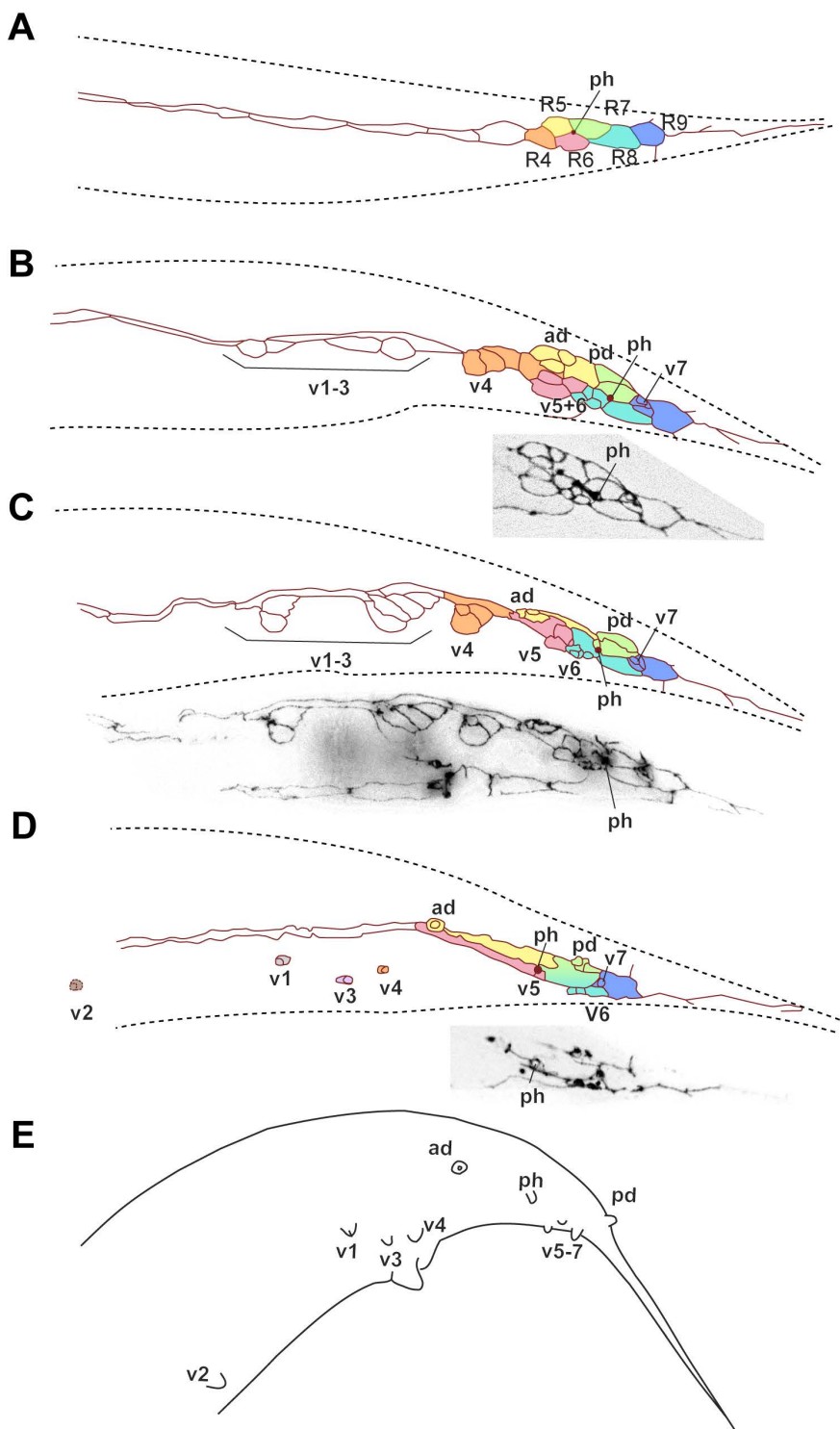

**Fig 9. MH27 antibody staining of adherens junctions in the tail of male *D. nasuensis* SB335 larvae.** Reconstructions of the adherens junction patterns (top) and part of the z-projected photomicrographs on which it is based (bottom) for **(A, B)** two L3 males, **(C)** a stage 1 male, and **(D)** a stage 3 male. **(E)** Schematic drawing of an adult male, left side. Raw image data are provided in S25–S28 Figs.

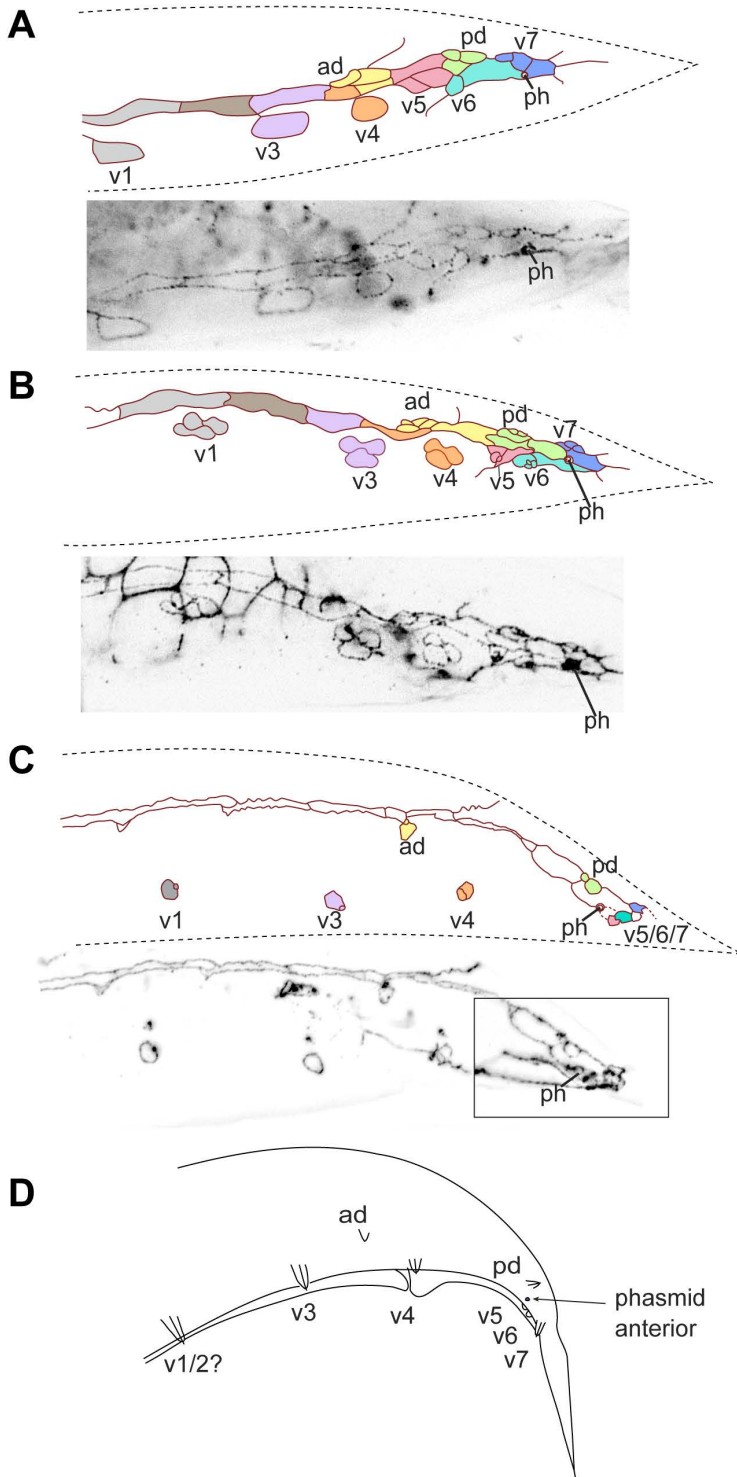

**Fig 10. MH27 antibody staining of adherens junctions in the tail of male *B. saprophaga* SB261 larvae.** Reconstructions of the adherens junction patterns (top) and part of the z-projected photomicrographs on which it is based (bottom) for **(A)** an early male, **(B)** a stage 1 male and **(C)** a stage 3 male (stages as in [11]). **(D)** Schematic drawing of an adult male. In **C**, the box shows a separate set of merged z-slices. In early males, the phasmid opening is located anterior of all GP groups. By stage 3, it is seen anterior of the three most posterior ray groups v5, 6 and 7. This is where it is found in adult males and **(D)** a stage 3 male. **(E)** Schematic drawing of an adult male. The phasmid opening is positioned anterior of the three posterior GP precursor cells before GP development begins. Raw image data are provided in S29–S31 Figs.

with the archetypal arrangement pattern [11]. To confirm that this approach holds for all rhabditids, we analysed the adherens junction patterns in developing males of three additional Rhabditina species and one representative of the outgroup.

The GP pattern in early L4 males of all species is in principle compatible with the archetype [11,21,28]. There are two main aspects to this archetype: First, the R*n* cells are produced by the cell lineage in a row in the lateral hypodermis (in the order R1, R2, R3, R4, R6, R5, R7, R8, R9). Second, anlagen for three GPs are positioned further dorsal than the others: R5 (producing ad), R7 (pd) and R9 (v7). GP v7 is positioned only slightly dorsal to the other "v" GPs, but this position is consistently seen during GP development and in adults of species without a bursa. (The bursa forces all GPs into a plane, and dorsoventral differences are restricted to the place where the tips of the GPs are anchored.) v7 is positioned most posterior in adults of all observed species except for *H. subulatum,* in which the anlagen for v5 and v6 shift their position past that of v7 such that v6 becomes the most posterior GP in this species (Fig 8). There is no evidence that any of the other "v" GPs have changed position relative to each other in *H. subulatum, P. oxycercus* and *B. saprophaga.* However, we found an exception in the diplogastrid *D. nasuensis.*

Diplogastrids are exceptional in that they have three instead of two dorsal GPs. The third dorsal GP lies anterior to the "ad" and "pd" GPs, and is the second or third GP along the anteroposterior axis. Its homology had remained an open question [14]. Our ablation results in *D. nasuensis* show that this additional dorsal GP is v1, derived from the V5 blast cell. In *D. nasuensis,* it is the second GP counted from anterior. Thus, in addition to being unusually far dorsal, v1 swapped ordinal position with v2, a situation not observed in any other species. We assumed that such a swap might be facilitated by a dorsally positioned v1 anlage and should be detectable by changes of AJ patterns. However, our MH27 stainings of *D. nasuensis* males were inconclusive. Division of the anterior R*n* cells is delayed relative to that of the posterior cells, which made it difficult to distinguish the anterior blast cells from lateral seam cells early during development. Later, the position and identity of the first three GP cell groups could also not be unequivocally distinguished by MH27 staining alone. The second and third GP cell groups were stacked in a dorsoventral manner in males at the middle of GP development. We did not obtain a sample in which this arrangement was resolved to match the arrangement of GPs in adults. The anlagen for v4-7, ad and pd are arranged in positions similar to those in other species and adheres to the archetypal pattern.

Although the archetypal pattern itself is highly conserved, it is established at different times during GP development: In *P. oxycercus* and *D. nasuensis* the R*n* cells are immediately reorganized such that R4, R6 and R8 are positioned further ventral than R5 R7 and R9. In *H. subulatum,* only R6 is positioned further ventral than all other R*n* cells. In *C. elegans*, the R*n* cells stay in a row after being produced, but after the first division of these cells, the R5 and R8 progeny are positioned further ventrally and the archetypal pattern is discernible. The same is true for *P. strongyloides* (Fig 11). In *B. saprophaga,* R1.a, R3.a and R4.a, which will form v1, v3 and v4, separate from the associated R*n*.p and move ventrally early but the v5 cell group becomes positioned ventral of the other GP cell groups only when the cells undergo the second division. By early L4 when all GP cell group divisions are completed, the archetypal pattern is visible in all species. Thus, the archetypal pattern can be achieved in different ways. When it is established early as in *P. oxycercus,* it may be the result of slightly modified division orientations of the parents of the R*n* cells. For instance, a dorsoventral or oblique division of V6.pppp would position R6 ventral of R5. Later, establishment of the pattern must be due to specific changes in cell contacts as the cells divide.

A GP arrangement with two postcloacal dorsal GPs and at least 5 ventral GPs is present in the stem species of Rhabditina (a.k.a. Clade V [40]), in representatives of Clade IV and in *Myolaimus* and *Deleya* species [41], which comprise the sister group of Clades IV + V. This conserved aspect of GP pattern is also discernible in some Spirurina [12]. Our analysis shows that the development of this GP pattern via the archetypal stage during early L4 is homologous in Rhabditina and outgroup representative Brevibuccidae. It is thus likely that the same constraints which lead to the archetypal pattern of GP anlage are present in all Rhabditida (Secernentea).

 

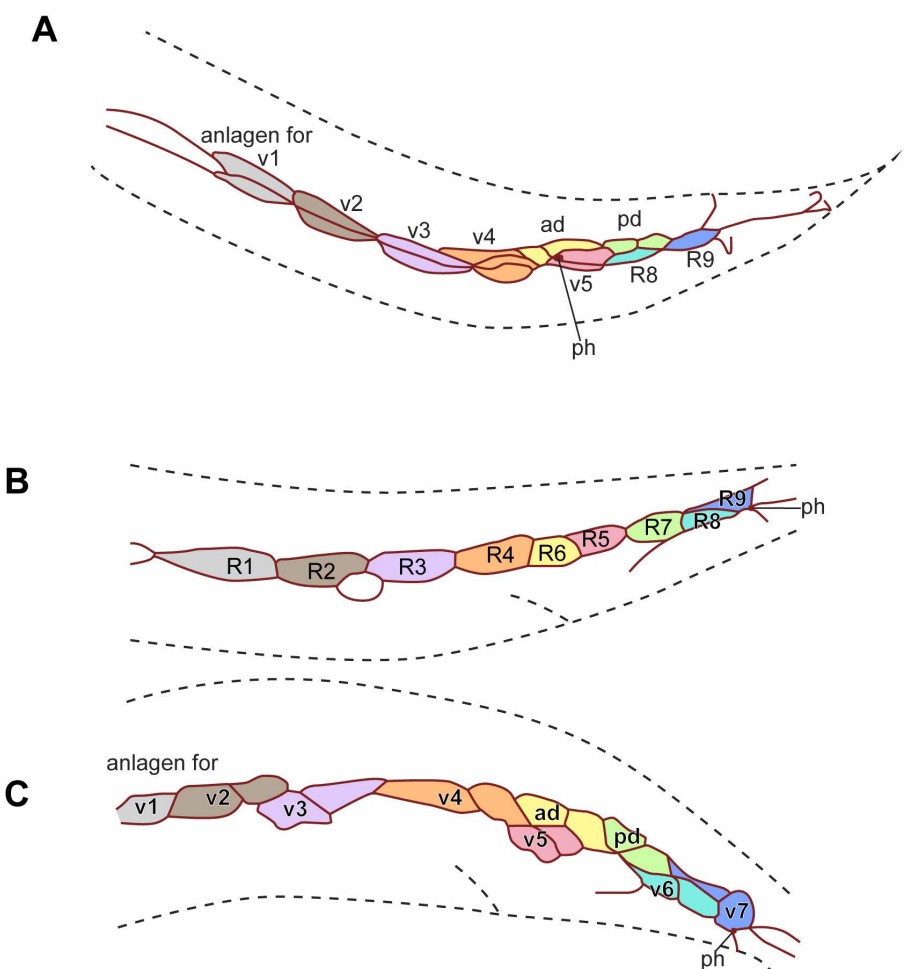

**Fig 11. Reconstruction of the AJ pattern of *Pelodera strongyloides* and *C. elegans* males early during GP development. (A)** *P. strongyloides* after the first division of most R*n* cells [11]. **(B, C)** Drawing after the *ajm-1*::GFP images from WormAtlas [39] for *C. elegans* males before **(A)** and after **(B)** the first division of the R*n* cells.

What might be the developmental basis for these constraints? As noted above, there are two main aspects to the archetype: the anteroposterior arrangement of R*n* precursors and a dorsoventral segregation of R5, R7 and R9 anlagen from the other GP anlagen. Analyses in *C. elegans* may provide insight into the developmental mechanisms that produce these patterns. First, the cell lineage involves a series of asymmetric divisions along the A-P (anteroposterior) axis. This axis is also patterned by HOX and other "selector" transcription-factor genes that specify different morphological and neurotransmitter fates for the cells that will make the GPs [42,43]. Second, the D-V (dorsoventral) axial patterning system that uses TGF-beta signaling further specifies the fates of GPs produced from R5, R7 and R9, with respect to both their dorsal position and the neurotransmitters [43,44]. Specifically, A-type neurons in the dorsal GPs (ad, pd and v7) are dopaminergic [44], whereas GPs derived from ventral anlagen (v1–v6) all use acetylcholine (ACh) as their primary neurotransmitter [45]. Mutations affecting either axial patterning system cause transformations in the GP morphological, positional and neurotransmitter fates. For example, loss-of-function mutations in the TGF-beta pathway cause the otherwise dorsal rays to position ventrally, resulting in fusions with ventral rays and loss of dopaminergic function [43,44]. We predict that these axial patterning systems are conserved among all species by stabilizing selection, due to deleterious

pleiotropic effects of mutations in those pathways, resulting in the conserved archetypal pattern of GP anlagen along the A-P and D-V axes.

## GP numbers

We next attempted to determine which GPs are absent in species with only 8 GP pairs: For *C. tripartitum,* laser ablations of cells in the T lineage led to the absence of only two GPs, pd and one of either v6 or v7 (which could not be distinguished as AJ staining for this species did not work). Similarly, in *M. blumi* v6 is absent because R8 does not divide [11]. In *P. oxycercus*, on the other hand, ablation of cells in the T lineage lead to the absence of three posterior GPs, showing that its missing GP is not derived from T. Instead, AJ stainings showed a conspicuous hypodermal cell between R1.p and the next R*n*.p cell (brown-colored cell in Fig 7B,C). We interpret this hypodermal cell as the undivided R2. Thus, in this species, v2 is most likely absent. A very similar AJ pattern is seen in *B. saprophaga.* Here, too, a hypodermal cell is located between the R*n*.p cell of the first and second GP cell group (brown-colored cell in Fig 10A,B). Again, we assume that R2 does not divide and that the missing GP is most likely v2. Thus, there are at least two different ways in which 8 GPs occur: lack of v2 or v6 (or possibly v7)*.*

As mentioned above, two interpretations are possible for the number of GPs in the Rhabditina stem species, either 8 or 9 pairs. That R2 does not divide in both *B. saprophaga* and *P. oxycercus* suggests that the v2 homolog is missing in both species, consistent with absence of v2 being a symplesiomorphic (shared ancestral) state in these species.

Assuming this scenario that the stem species of Rhabditina had 8 GPs, one GP would then have been gained in both the lineage to Bunonematidae and in rhabditids after the divergence of *Poikilolaimus*. According to our interpretation, the gained GP is most likely v2 since this GP is present in nearly all Rhabditina species derived after the divergence of *Poikilolaimus*. For Bunonematidae, neither the cell lineage nor AJ pattern is known. However, Fürst von Lieven [46] compared the GP pattern of *Bunonema pini* with that of diplogastrids and found them to be similar. It is therefore unlikely that a GP in the posterior part of the tail was added in this lineage. It is more likely that, as in rhabditids, the homolog of R2 divided to produce the additional GP. Alternatively, this GP in Bunonematidae could be derived from the more anterior blast cell V5.

Gain of GPs in the anterior part of the tail appears to be developmentally easy to achieve. There are several reports of aberrations with ectopic GPs, always anterior of the most anterior GP in a species (e.g., *Pelodera cutanea* [47], *M. blumi* [11], *Steinernema bibionis* [12], *Pellioditis pelhamensis* [48]). We observed a *P. oxycercus* male with two ectopic GPs anterior of v1 on one side of the tail (Figs 12, S32 Fig). Nevertheless, within Rhabditina and most lineages of Clade IV, the number of GPs is restricted to nine pairs or fewer (*Steinernema* is an exception). It is possible that for Rhabditina, an optimal number of GPs lies between 7 and 9 and that variants with more GPs are selected against. That is, as variation causing >9 GPs exists naturally, selection appears to constrain GP number.

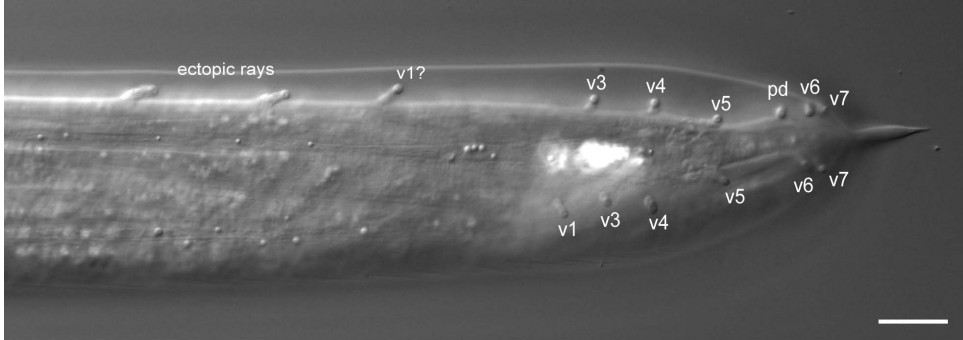

**Fig 12. Photomicrograph of a *P. oxycercus* male with two ectopic GPs on the left body side.** Raw image data are provided in S32 Fig.

Why most species have nine GPs is a matter of speculation. For example, males with more GPs may incur an energetic cost of producing or maintaining extra, redundant organs. Genetic and ablation studies have demonstrated considerable redundancy among rays in *C. elegans* for certain aspects of male mating behavior. As few as three pairs of rays are required to allow successful mating in laboratory experiments [49]. However, mating robustness is enhanced with additional rays, with at least 7 GPs needed for optimal mating [49,50]. According to a recent neurotransmitter "atlas" for *C. elegans* males [45], there is a unique neurotransmitter combination expressed by the A and B-type neurons for each left-right GP pair—except that v1 is redundant with v3 and v2 is redundant with v5. Thus, one constraint may involve maintaining the seven posterior GPs, which are functionally nonredundant with respect to neurotransmitters.

Within Rhabditina, however, the GP that was lost in two independent lineages (*Metarhabditis* and *Cruznema)* was a ventral GP derived from the T blast cell, possibly v6 in both cases. Fitch [28] noted that in *C. elegans*, expression of the gene *lin-32* is required for GP formation, and mutation in this gene leads to the absence of GPs. Weak alleles of *lin-32* affect v6 more than other GPs, constituting a potential bias or constraint [51]. Thus, variation in expression of the *lin-32* ortholog or other genes in the neurogenic pathway could potentially explain the recurrent losses of v6.

## Phasmid position

To evaluate the hypotheses for how the phasmid position anterior of the posterior-most 3 GP develops—by migration or polarity reversal of the first T-cell division (Fig 1), we ablated T and its progeny in three species with anterior phasmids, *P. strongyloides, C. tripartitum* and *D. nasuensis.* The polarity reversal hypothesis posits that in these species, the anterior daughter of T (T.a) should make the phasmid socket cell, and the posterior daughter T.p should make 3 GPs. Our ablation results refute this hypothesis. In all three species, ablation of T.p or T.pa led to the absence of the phasmid, and ablation of T.a or T.ap led to the absence of GPs. This suggests that the T lineage is the same as in *C. elegans* and *P. redivivus,* and changes in the phasmid position are due to cell migration and not reversal in T cell polarity.

Fitch & Emmons [11] documented several migration events of GP cell groups during GP development in the late L3/early L4 stage. It was thus conceivable that the change in phasmid position might occur in the context of these cell migrations. We scrutinized MH27-stained males of species with posterior phasmids early during GP development and determined the position of the phasmids at this time. In two Rhabditina species with anterior phasmids, *H. subulatum* and *D. nasuensis*, the phasmid is already located anterior of the R7 blast cell before division of the R*n* cells (Figs 8, 9). We re-analyzed photographs for *P. strongyloides* taken during the initial study of GP development [11] and found that here too the phasmid is located anterior of R7 (Fig 11). Thus, in these species, the phasmid socket and the precursors of the three posterior-most GPs swap ordinal position before GP development begins. If this positional change happened before the division of T.ap, which gives rise to R7, R8 and R9, it would also explain why the phasmid opening is anterior of exactly three GPs in almost all species with anterior phasmids. The arrangement of T descendants in *D. nasuensis* suggests how this change in position may happen: T.pa, which makes the phasmid socket, is already located slightly anterior (and dorsal) to T.ap early in L2. The picture is different in *B. saprophaga*: In the earliest male for which we could analyze the AJ patern, the phasmid was located at the border between R8.p and R9.p, almost posterior to all GP cell groups (Fig 10A), consistent with its origin from the posterior daughter of T. Later the v5, v6 and v7 cell groups shift or migrate posteriorly, resulting in the anterior phasmid position seen in adults.

We conclude that even though the phasmid socket cell is derived from T.p in all species tested, anterior phasmids can develop by cell migration at different times: early, before R7-R9 are born as in *D. nasuensis,* or late during L4 as in *B. saprophaga*. That is, there is likely to be a tight constraint on the cell lineage, in particular the polarity of the division of the T blast cell. One possible reason for such a constraint is that variation in a pleiotropically employed signaling pathway like Wnt, which is responsible for the T blast-cell polarity [32], would deleteriously impact embryogenesis and other developmental fate decisions and thus would be selected against.

In summary, we set out to determine the degree to which the recurrent evolution of morphological characters in the rhabditid nematode male tail arose by changes to the same developmental events. We find that, despite interspecific diversity in final GP or phasmid positions, the archetypal pattern of anlagen along the A-P and D-V axes is highly conserved, as is the orientation of the divisions in the cell lineage that we tested by laser ablation. We speculate that the main constraints on evolution of GP and phasmid morphology, function and position involve stabilizing selection on the A-P, D-V and cell polarity patterning systems. Prior developmental events can be variable, as long as the archetypal stage appears at some point. Subsequently, other processes that presumably can be tuned more specifically to particular tissues, like cell migratory behaviors, then provide the means to evolve diversity in GP/phasmid positions. There also appears to be substantial variation allowed in the timing at which migration could happen. Finally, although 9 GPs is typical for rhabditids, reduction to 8 can become fixed in some species. Functional redundancy (e.g., with respect to neurotransmitter identity) may influence which GPs are more or less likely to be affected, consistent with redundancy allowing some degree of evolvability.

## Methods

### Animal husbandry

*P. strongyloides* and *P. oxycercus* were grown on NGM plates seeded with *E. coli* strain OP50−1. *C. tripartitum* and *D. nasuensis* were grown on plates containing 1% agar in tap water supplemented with a small piece of NGM agar and bacteria originating from the original habitat.

### Laser ablations

Laser ablation of blast cells in the male tail was performed as described previously [52] using a Zeiss Axioskop I with a 337-nm nitrogen pumping laser (Micropoint, Photonic Instruments) connected via optical cable to a dye cell module at the epiillumination port filled with coumarin 440 dye (5 mM in methanol). Targeting was performed using a video camera connected to a TV monitor. Ablated cells were documented by drawings using a camera lucida attached to the microscope and the resulting adult phenotype with photographs. Growing conditions and conditions for ablations had to be adjusted for each species, as follows.

**P. strongyloides strain DF5022.** Gravid females were treated with alkaline hypochlorite solution (3 volumes household bleach, 2 volumes 4 M NaOH) to release embryos, which were kept in M9 buffer for about 24h (not longer) and were then placed on NGM plates seeded with OP50 for 10 h at 20 ºC. At this time, the T cell had divided once in most animals. The T daughters divided 12 h after feeding. For ablations, L1 worms were placed in 2 µl M9 with 20 mM sodium azide on a pad prepared with 5% Difco Noble agar in water. After ablation, worms were recovered from the slide and placed onto individual NGM plates seeded with OP50−1. After they developed into adults, they were mounted again to assess the effect of the ablation.

Before the division of their T cells, several L1s were placed into 4 µl M9 buffer and some bacteria on a pad of 5% agar and covered with a coverslip that was then sealed with vacuum grease. Development of these worms was observed until the T cell had divided.

**C. tripartitum strain SB202.** As this species does not survive bleaching, gravid adults were allowed to lay eggs for a few hours. Embryogenesis took more than 48 hours at 20 ºC. After that time, freshly hatched L1s were collected onto new plates. The T cell divided approximately 16 hours after hatching, at which time ablations were performed.

To immobilize the L1 for ablations, they were placed in 2 µl water onto 5% water agar pads that contained 10 µM sodium azide. After ablations, the worms were recovered onto individual water agar plates supplied with some food bacteria taken from the culture plates.

***D. nasuensis* strain DF335.** Since diplogastrids undergo the first larval stage inside of the eggshell, we investigated newly hatched L2 larvae that were picked from culture plates and transferred to a slide with an agar pad containing 20 μM sodium azide for ablations. After ablations, the worms were recovered onto individual water agar plates supplied with some food bacteria taken from the culture plates.

## MH27 staining

As described previously [11], MH27 mouse monoclonal antibody (from supernatant of cells provided by R. Waterston) was used with rhodamine-conjugated goat anti-mouse secondary antibody (Sigma-Aldrich) to stain animals fixed with paraformaldehyde. Cultures were synchronized by bleaching or hatch-off as described above, and animals were collected when most were at the desired stage (L3-L4) as assessed by observation using a stereo dissecting microscope. The stained specimens were mounted on slides in SlowFade covered with a cover slip and the edges sealed with nail polish. Rhodamine fluorescence was observed with the Zeiss Axioskop I and z-stack images captured using a Hamamatsu C4742-95 "Orca" camera and OpenLab software (ver. 3.0.9, Improvision). Some z-stacks were also deconvolved using OpenLab.

## Image processing

Files in the.liff format were imported to FIJI (Image J) and processed with this software.

Images were inverted (Edit>invert) and the background removed (Process>Subtract background: light background, rolling ball radius 10 pixels). Z projection (Image>Stack>z project: min setting or sum setting) was used to merge several slices. Multiple merges were then used to reconstruct the patterns of adherens junctions in the tail of males at different developmental times during the L4 stage.

## Phylogenetic analysis

Using the cladogram published earlier for rhabditids [31], we traced character evolution using the parsimony method [53,54], checking our inferences with Mesquite [55].

## Supporting information

**S1 Fig. Raw image data for Fig 3A.** *Pelodera strongyloides* strain DF5022, L1 larva, 0h post-hatch (sex unknown). DIC micrographs as a z-stack, TIFF format.
(TIF)

**S2 Fig. Raw image data for Fig 3B.** *Pelodera strongyloides* strain DF5022, L1 larva, 13h post-hatch (sex unknown). DIC micrographs as a z-stack, TIFF format.
(TIF)

**S3 Fig. Raw image data for Fig 3D.** *Pelodera strongyloides* strain DF5022, adult male, ablation result #44 (see Table 1). DIC micrographs as a z-stack, TIFF format.
(TIF)

**S4 Fig. Raw image data for Fig 3E.** *Pelodera strongyloides* strain DF5022, adult male, ablation result #46 (see Table 1). DIC micrographs as a z-stack, TIFF format.
(TIF)

**S5 Fig. Raw image data for Fig 3F.** *Pelodera strongyloides* strain DF5022, adult male, ablation result #13 (see Table 1). DIC micrographs as a z-stack, TIFF format.
(TIF)

**S6 Fig. Raw image data for Fig 4A.** *Cruznema tripartitum* strain SB202, male L1 larva used for ablation #17 (see Table 1). DIC micrographs as a z-stack, TIFF format.
(TIF)

**S7 Fig. Raw image data for Fig 4B.** *Cruznema tripartitum* strain SB202, L1 larva after first division of T (animal #33). DIC micrographs as a z-stack, TIFF format.
(TIF)

**S8 Fig. Raw image data for Fig 4C.** *Cruznema tripartitum* strain SB202, larval female used in ablation #35 (see Table 1). DIC micrographs as a z-stack, TIFF format.
(TIF)

**S9 Fig. Raw image data for Fig 4E.** *Cruznema tripartitum* strain SB202, adult male, ablation result #17 (see Table 1). DIC micrographs as a z-stack, TIFF format.
(TIF)

**S10 Fig. Raw image data for Fig 4F.** *Cruznema tripartitum* strain SB202, adult male, ablation result #31 (see Table 1). DIC micrographs as a z-stack, TIFF format.
(TIF)

**S11 Fig. Raw image data for Fig 4G.** *Cruznema tripartitum* strain SB202, adult male, ablation result #25 (see Table 1). DIC micrographs as a z-stack, TIFF format.
(TIF)

**S12 Fig. Raw image data for Fig 5A.** *Diplogasteroides nasuensis* strain SB335, L2 female for ablation #1 (see Table 1). DIC micrographs as a z-stack, TIFF format.
(TIF)

**S13 Fig. Raw image data for Fig 5C.** *Diplogasteroides nasuensis* strain SB335, adult male ablation result #36 (see Table 1). DIC micrographs as a z-stack, TIFF format.
(TIF)

**S14 Fig. Raw image data for Fig 5D.** *Diplogasteroides nasuensis* strain SB335, adult male ablation result #18 (see Table 1). DIC micrograph.
(TIF)

**S15 Fig. Raw image data for Fig 6A.** *Poikilolaimus oxycercus* strain EUK103 L1 larva after the first division of T. DIC micrographs as a z-stack, TIFF format.
(TIF)

**S16 Fig. Raw image data for Fig 6B.** *Poikilolaimus oxycercus* strain EUK103 adult male after ablation of T.a. DIC micrographs as a z-stack, TIFF format.
(TIF)

**S17 Fig. Raw image data for Fig 7A.** *Poikilolaimus oxycercus* strain EUK106 early L3 male, immunofluorescently stained with MH27 antibody. Epifluorescence micrographs as a z-stack, TIFF format.
(TIF)

**S18 Fig. Raw image data for Fig 7B.** *Poikilolaimus oxycercus* strain EUK106 L3 male, immunofluorescently stained with MH27 antibody. Epifluorescence micrographs as a z-stack, TIFF format.
(TIF)

**S19 Fig. Raw image data for Fig 7C.** *Poikilolaimus oxycercus* strain EUK106 L4 (stage 1) male, immunofluorescently stained with MH27 antibody. Epifluorescence micrographs as a z-stack, TIFF format.
(TIF)

**S20 Fig. Raw image data for Fig 7D.** *Poikilolaimus oxycercus* strain EUK106 L4 (stage 3) male, immunofluorescently stained with MH27 antibody. Epifluorescence micrographs as a z-stack, TIFF format.
(TIF)

**S21 Fig. Raw image data for Fig 8A.** *Haematozoon subulatum* strain SB303 early L3 male, immunofluorescently stained with MH27 antibody. Epifluorescence micrographs as a z-stack, TIFF format.
(TIF)

**S22 Fig. Raw image data for Fig 8B.** *Haematozoon subulatum* strain SB303 L3 male, immunofluorescently stained with MH27 antibody. Epifluorescence micrographs as a z-stack, TIFF format.
(TIF)

**S23 Fig. Raw image data for Fig 8C.** *Haematozoon subulatum* strain SB303 L4 (stage 1) male, immunofluorescently stained with MH27 antibody. Epifluorescence micrographs as a z-stack, TIFF format.
(TIFF)

**S24 Fig. Raw image data for Fig 8D.** *Haematozoon subulatum* strain SB303 L4 (stage 3) male, immunofluorescently stained with MH27 antibody. Epifluorescence micrographs as a z-stack, TIFF format.
(TIF)

**S25 Fig. Raw image data for Fig 9A.** *Diplogasteroides nasuensis* strain SB335 L3 male, immunofluorescently stained with MH27 antibody. Epifluorescence micrographs as a z-stack, TIFF format.
(TIF)

**S26 Fig. Raw image data for Fig 9B.** *Diplogasteroides nasuensis* strain SB335 L3 male, immunofluorescently stained with MH27 antibody. Epifluorescence micrographs as a z-stack, TIFF format.
(TIF)

**S27 Fig. Raw image data for Fig 9C.** *Diplogasteroides nasuensis* strain SB335 L4 (stage 1) male, immunofluorescently stained with MH27 antibody. Epifluorescence micrographs as a z-stack, TIFF format.
(TIF)

**S28 Fig. Raw image data for Fig 9D.** *Diplogasteroides nasuensis* strain SB335 L4 (stage 3) male, immunofluorescently stained with MH27 antibody. Epifluorescence micrographs as a z-stack, TIFF format.
(TIF)

**S29 Fig. Raw image data for Fig 10A.** *Brevibucca saprophaga* strain SB261 L3 male, immunofluorescently stained with MH27 antibody. Epifluorescence micrographs as a z-stack, TIFF format.
(TIF)

**S30 Fig. Raw image data for Fig 10B.** *Brevibucca saprophaga* strain SB261 L4 (stage 1) male, immunofluorescently stained with MH27 antibody. Epifluorescence micrographs as a z-stack, TIFF format.
(TIF)

**S31 Fig. Raw image data for Fig 10C.** *Brevibucca saprophaga* strain SB261 L4 (stage 3) male, immunofluorescently stained with MH27 antibody. Epifluorescence micrographs as a z-stack, TIFF format.
(TIF)

**S32 Fig. Raw image data for Fig 12.** *Poikilolaimus oxycercus* strain EUK103 adult male with ectopic GPs on the left side. DIC micrographs as a z-stack, TIFF format.
(TIF)

## Author contributions

**Conceptualization:** David H. A. Fitch.

**Data curation:** Karin Kiontke.

**Funding acquisition:** David H. A. Fitch.

**Investigation:** Karin Kiontke, Simone Kolysh, Rocio Ng, David H. A. Fitch.

**Resources:** Karin Kiontke.

**Supervision:** David H. A. Fitch.

**Validation:** Karin Kiontke, Simone Kolysh, Rocio Ng.

**Visualization:** Karin Kiontke, Simone Kolysh, Rocio Ng.

**Writing – original draft:** Karin Kiontke.

**Writing – review & editing:** Karin Kiontke, Simone Kolysh, Rocio Ng, David H. A. Fitch.

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
