## [Decision Letter · Decision Letter 0]

18 Feb 2026

Dear Dr. Fitch,

Thank you for submitting your manuscript to PLOS ONE. After careful consideration, we feel that it has merit but does not fully meet PLOS ONE’s publication criteria as it currently stands. Therefore, we invite you to submit a revised version of the manuscript that addresses the points raised during the review process.

We look forward to receiving your revised manuscript.

Kind regards,

Ebrahim Shokoohi

Academic Editor

PLOS One

Journal Requirements:

“This work was funded by grants to DHAF from the National Institutes of Health (R01-GM141395) and the National Science Foundation (IOB-0643047). SK and RN were supported by Deans Undergraduate Research Fellowships from the College of Arts and Science at New York University.”

“This work was funded by grants to DHAF from the National Institutes of Health (R01 GM141395) and the National Science Foundation (IOB-0643047). SK and RN were supported by Deans Undergraduate Research Fellowships from the College of Arts and Science at New York University.”

“This work was funded by grants to DHAF from the National Institutes of Health (R01-GM141395) and the National Science Foundation (IOB-0643047). SK and RN were supported by Deans Undergraduate Research Fellowships from the College of Arts and Science at New York University.”

**Additional Editor Comments:**

Dear Authors,

Thank you for submitting your manuscript entitled “Developmental Constraints in the Repeated Evolution of Male Tail Characters in Rhabditid and Diplogastrid Nematodes.” The topic is highly relevant to evolutionary developmental biology of nematodes, and the comparative framework across Rhabditidae and Diplogastridae has the potential to make a meaningful contribution to our understanding of morphological constraint and character evolution.

While the study is promising, several substantive issues need to be addressed to strengthen the conceptual framework, methodological transparency, and robustness of the conclusions.

The referees commnets are given for your reference.

Reviewers' comments:

Reviewer's Responses to Questions

**Comments to the Author**

1. Is the manuscript technically sound, and do the data support the conclusions?

Reviewer #1: Yes

Reviewer #2: Yes

2. Has the statistical analysis been performed appropriately and rigorously?

Reviewer #1: Yes

Reviewer #2: N/A

3. Have the authors made all data underlying the findings in their manuscript fully available?

Reviewer #1: No

Reviewer #2: Yes

4. Is the manuscript presented in an intelligible fashion and written in standard English?

Reviewer #1: Yes

Reviewer #2: Yes

Reviewer #1: Title: Developmental constraints in the repeated evolution of male tail characters in rhabditid and diplogastrid nematodes

General Assessment

This manuscript presents a detailed evo-devo analysis of recurrently evolving male tail characters in rhabditid and diplogastrid nematodes. By combining phylogenetic mapping, adherens junction (MH27) staining, and laser ablation experiments, the authors provide a rich and carefully executed dataset. The study addresses a classic and important question in evolutionary biology—the role of developmental constraints in shaping morphological evolution—and is well aligned with the scope of PLOS ONE.

Overall, the experimental work is strong and the manuscript represents a valuable contribution. However, several conceptual and interpretative issues need to be addressed before the conclusions can be fully supported.

Major Comments

1. Clarification of “developmental constraint”

The manuscript repeatedly invokes developmental constraint, but the distinction between constraint, developmental bias, canalization, and parallelism is not always clear.

The authors should provide a more explicit operational definition of what constitutes evidence for developmental constraint in this study.

Not all cases of parallel or repeated evolution necessarily imply strong constraint; in some instances, the data may also be consistent with limited developmental flexibility.

A clearer conceptual framing, particularly in the Introduction and early Discussion, would strengthen the interpretation of the results.

2. Dependence on phylogenetic assumptions

The evolutionary reconstructions rely heavily on a specific phylogenetic topology (Kiontke et al. 2011), despite acknowledged differences with more recent genome-based phylogenies.

Several key conclusions (e.g., ancestral number of genital papillae and gain vs. loss scenarios) are sensitive to tree topology.

The authors should more explicitly discuss how alternative phylogenetic placements (e.g., of Diplogastridae) could affect their inferences.

The use of Dollo-like assumptions (DELTRAN) should be better justified, and its limitations more clearly acknowledged.

3. Homology inference of genital papillae

The archetypal GP pattern is a powerful framework, but in some taxa (notably Diplogasteroides nasuensis and Brevibucca saprophaga), homology assignments rely on indirect inference.

The degree of uncertainty in these homology assignments should be stated more explicitly.

Where appropriate, alternative interpretations should be acknowledged, even if considered less likely.

4. Interpretation of laser ablation results

The laser ablation experiments are a major strength of the study, but in some species the number of informative ablations is limited.

Conclusions such as the statement that phasmid position changes are never due to lineage changes may be overly strong.

The authors should adopt a more cautious phrasing (e.g., “our data suggest”) and explicitly note experimental limitations, particularly in taxa with small sample sizes.

Minor Comments

Clarity and structure

Some Results sections are very dense and would benefit from shorter paragraphs or clearer sub-structuring.

Figures

Figures 7–10 are information-rich but complex; additional simplified schematics could help readers unfamiliar with nematode tail anatomy.

Terminology

The Pel/Lep terminology could be introduced more succinctly earlier in the manuscript.

Data availability

Please clarify whether raw image stacks from MH27 staining will be made publicly available.

Reviewer #2: The manuscript described the Developmental constraints in the repeated evolution of male tail characters in rhabditid and diplogastrid nematodes. The article is well-written but it is not acceptable for publication in the current format, however, it is acceptable after a minor revision. The following issues should be addressed by the authors.

Specific comments:

1. Kindly include molecular analysis subsection under methodologies to the article.

2. How many organisms or samples were studied for this investigation?

3. Kindly add conclusion section to the article.

4. What is the reason behind choosing this topic for research?

5. Kindly elaborate your hypothesis in the article.

6. Kindly include the abbreviations meaning in every image titles.

7. In Fig. 6(B), what’s the *stands for?

8. Which molecular markers or characters have been used to deduce the phylogenetic tree?

.

Reviewer #1: No

Reviewer #2: **Yes:** Himani SharmaHimani SharmaHimani SharmaHimani Sharma

---

## [Author Response · Author response to Decision Letter 1]

6 Apr 2026

Please see the Revised Manuscript, the Markup (Track-Changes) version, and the detailed Response to Reviewers.

---

## [Editor Report · Decision Letter 1]

13 Apr 2026

Developmental constraints in the repeated evolution of male tail characters in rhabditid and diplogastrid nematodes

PONE-D-26-00083R1

Dear Dr. Fitch,

We’re pleased to inform you that your manuscript has been judged scientifically suitable for publication and will be formally accepted for publication once it meets all outstanding technical requirements.

Kind regards,

Ebrahim Shokoohi

Academic Editor

PLOS One

Additional Editor Comments (optional):

The authors have adequately addressed the reviewers’ concerns and revised the manuscript accordingly, resulting in an improved version of the paper.
---

## [Editor Report · Acceptance letter]

PONE-D-26-00083R1

PLOS One

Dear Dr. Fitch,

I'm pleased to inform you that your manuscript has been deemed suitable for publication in PLOS One. Congratulations! Your manuscript is now being handed over to our production team.

Kind regards,

on behalf of

Dr. Ebrahim Shokoohi

Academic Editor

PLOS One